# Textual Supervision Enhances Geospatial Representations in Vision-Language Models

## Abstract

Geospatial understanding is a critical yet underexplored dimension in the development of machine learning systems for tasks such as image geolocation and spatial reasoning. In this work, we analyze the geospatial representations acquired by three model families: vision-only architectures (e.g., ViT), vision-language models (e.g., CLIP), and large-scale multimodal foundation models (e.g., LLaVA, Qwen, and Gemma). By evaluating across image clusters, including people, landmarks, and everyday objects, grouped based on the degree of localizability, we reveal systematic gaps in spatial accuracy and show that textual supervision enhances the learning of geospatial representations. Our findings suggest the role of language as an effective complementary modality for encoding spatial context and multimodal learning as a key direction for advancing geospatial AI.

## 1 Introduction

Vision models have undergone tremendous progress in the last decade, driven by advances in convolutional neural network (CNN) architectures (Simonyan & Zisserman, 2014; He et al., 2016) and Vision Transformers (ViT) (Dosovitskiy et al., 2021). These models are capable of capturing high-level, transferable representations that can be utilized in zero-shot scenarios via their embeddings and be adapted through fine-tuning for specific downstream tasks. Specifically, ViTs benefit from the scalability of Transformers (Vaswani et al., 2017) and enable the development of foundation models across multiple data modalities and application domains.

Recent advances such as CLIP (Radford et al., 2021) include multimodal models that integrate text and vision to learn joint representations within a shared latent space. Another line of research focuses on vision-language models (VLMs), which integrate text and image inputs through a two-stage training pipeline: an initial phase using paired text-image data, followed by instruction tuning (Liu et al., 2024; Bai et al., 2025; Kamath et al., 2025). These models typically employ a frozen vision encoder alongside a language model, enabling multimodal understanding and generation.

Vision models increasingly demonstrate the ability to internalize diverse meta information around the world, raising the question of whether they also encode *geolocation*—even without explicit geospatial supervision. Their internal representations are shaped by architecture, pretraining, and fine-tuning, yet remain difficult to interpret (Ghiasi et al., 2022). This challenge is further amplified in emerging VLMs, where multimodal complexity obscures the mechanisms by which knowledge is encoded. Such opacity can lead to unintended outcomes, including geographic disparities (Moayeri et al., 2024). To improve fairness and transparency, we examine the capacity of vision-only and vision-language models to encode implicit geospatial information (Figure 1) by asking the following question: **To what extent do these models internalize global location knowledge during their training and fine-tuning pipelines?**

Here, we use the term *geospatial representations* to refer to latent features contained in the inner layers of ViTs or VLMs that encode information relevant to downstream geospatial tasks, i.e., related to physical location on Earth. Learning geospatial representations via supervised training has already been explored by Vivanco Cepeda et al. (2023). Following on their work, we are interested in investigating what kinds of geospatial features are learned during training without additional supervision. For text-based large language models (LLMs), Gurnee & Tegmark (2024) and Godey et al. (2024) have shown that specific neurons and layers within LLMs implicitly encode latitude and longitude information and that this capacity scales with increasing model size.

Figure 1: Schematic illustration of our linear probing analysis setup. We fit ridge regression models $\mathbf{W}$ on the [CLS] or last token residuals $\boldsymbol{A}^{(l)}$ from each layer $l$ to investigate whether geospatial information (i.e., latitude and longitude) can be extracted from these hidden dimensions based on the $R^2$ value.

Embeddings from pretrained LLMs can also be used to create geospatial embeddings from geolocation-related prompts, as shown in LLMGeovec (He et al., 2025), where its representations improve performance on various downstream tasks requiring geospatial understanding. Additionally, Roberts et al. (2024) has shown that VLMs have spatial reasoning capabilities, being able to complete a variety of tasks through zero-shot settings. We extend these results by focusing on how ViT models separate images spatially in their learned latent space and exploring how these models learn geospatial representations.

Our main contributions are as follows.

- We investigate geospatial representations learned by vision-only architectures, vision-language models, and large-scale multimodal foundation models, finding that the latter two groups exhibit substantially stronger geospatial structure.

- We evaluate the performance of different model representations using layer-wise probing for geospatial location prediction. We find that vision-only models tend to exhibit stronger representations on their last layer, while VLMs have better geospatial representations on the early layers of their language model block.

- We show that prompting VLMs allows the geospatial information to be propagated to the latter layers, in some cases leading to an increase in representation quality.

## 2 RELATED WORKS

### 2.1 VISION-BASED MODELS

ViTs emerged as a paradigm shift in computer vision, adapting Transformers (Vaswani et al., 2017) to image data by segmenting images into tokenized patches with positional embeddings (Dosovitskiy et al., 2021). The initial ViT model demonstrated that large-scale supervised pretraining on image classification tasks could yield transferable representations across domains. Researchers have also explored self-supervised approaches to vision. One such method is the Masked Autoencoder (MAE) (He et al., 2022), which trains models to reconstruct randomly masked image patches. This objective encourages the extraction of semantically rich features that can be effectively adapted to downstream tasks via fine-tuning.

As an alternative to self-supervised pretraining, Caron et al. (2021) proposed a self-distillation framework named DINO. This approach enables the model to learn invariant representations across multiple augmented views of the same image, resulting in linearly separable features that implicitly capture semantic structures such as object boundaries and regions. Building upon this foundation, DINOv2 (Oquab et al., 2023) extends this methodology by incorporating the IBOT loss (Zhou et al., 2021), a patch-level objective. This integration facilitates scalable pretraining, enhancing the model's capacity to learn visual representations from large-scale unlabeled datasets.

## 2.2 VISION-LANGUAGE MODELS

A turning point for vision models has been the integration of language for learning shared representations. Pioneering work like CLIP jointly trained a vision encoder and a language encoder to align their representations using a large corpus of web-scraped image-caption pairs (Radford et al., 2021). Subsequent research, such as SigLIP (Zhai et al., 2023), expanded CLIP by replacing softmax loss with a sigmoid objective, decoupling performance from batch size, and allowing for improved scalability. These models are fundamental and are applied as the core vision-encoder for many of the vision-language foundation models presented in our work.

Foundation VLMs utilize specialized training pipelines. For example, LLaVA-1.5 (Liu et al., 2024) employs a two-stage training process, first aligning a frozen CLIP visual encoder with an LLM on image-text pairs, and then fine-tuning the model on a GPT-generated instruction-following dataset to enhance its conversational and reasoning abilities. Qwen2.5 (Bai et al., 2025) follows a similar process, pretraining on image-text pairs and then performing additional supervised fine-tuning (SFT) and direct preference optimization (DPO) to structure instruction-following data. Gemma 3 (Kamath et al., 2025) leverages a frozen SigLIP vision encoder, a pretraining stage similar to previous models, and a post-training stage that includes knowledge distillation from a larger instruction-tuned model and alignment with human feedback via SFT and reinforcement learning with human feedback (RLHF).

## 3 METHODS

### 3.1 MODELS

To examine how geolocation capabilities emerge in vision models without explicit supervision, we curated a diverse set of architectures spanning multiple modalities and training paradigms. Our selection includes both (i) vision-only encoders, i.e., ViT (Dosovitskiy et al., 2021), ViT Masked Autoencoder (He et al., 2022), DINOv2 (Oquab et al., 2023), and Web-SSL models (Fan et al., 2025), and (ii) vision–language models, i.e., CLIP (Radford et al., 2021), MetaCLIP (Xu et al., 2024), LLaVA-1.5 (Liu et al., 2024), Qwen2.5 (Bai et al., 2025), and Gemma 3 (Kamath et al., 2025), representing supervised and self-supervised approaches. For each model family, we evaluated at least two size variants to assess the influence of scale on learned geospatial representations. Additional information about these models is given in Appendix A.

### 3.2 DATASET

To build our dataset, we sampled images from established benchmarks, including YFCC100M and Google Landmarks. We provide the details below.

**Yahoo Flickr Creative Commons 100 Million (YFCC100M) (Thomee et al., 2016).** We used a 4M-image subset from the MediaEval 2016 Placing Task competition (Choi et al., 2016), obtained via Kaggle[1]. This dataset is a diverse collection of Flickr-sourced images spanning natural scenes, urban environments, and everyday objects with location data. To analyze localizability across semantic categories, we partitioned this subset via unsupervised clustering. First, we extracted image embeddings with ResNet-152 (He et al., 2016) pretrained on ImageNet (Deng et al., 2009), then we applied principal component analysis (PCA) to retain the top-100 components explaining the highest variance and performed $k$-means clustering (Han et al., 2022). Among 19 tested $k$ values ($k = 10, 15, \ldots, 100$), we selected $k = 40$ using the elbow method (Thorndike, 1953), with manual inspection confirming semantic coherence. The resulting clusters captured meaningful categories, including people, objects, cliffs, landscapes, and buildings. Complete clustering details are provided in Appendix B.

**Google Landmarks (Weyand et al., 2020).** This dataset contains over 5M photographs of globally recognized landmarks and tourist sites (e.g., Eiffel Tower, Mount Fuji) that are available in public data sources. We hypothesize that such iconic scenes may have been encountered during model pretraining, contributing to their high localizability. It also contains non-localizable images, such as particularly close-up shots of people or animals, and generic textures such as soil, which lack

---

[1]https://www.kaggle.com/datasets/habedi/large-dataset-of-geotagged-images

distinctive geospatial cues. In our experiments, we use a subset of 580k images[2], hereafter referred to as the *Landmarks* dataset, with geolocation coordinates extracted from OpenStreetMap.

### 3.2.1 SAMPLING

Across all datasets, the geospatial distribution of images was imbalanced, skewed toward major cities in Europe and North America. To mitigate this bias, we partitioned the globe into non-overlapping geocells based on global administrative area boundaries and iteratively merged regions with insufficient samples. We merged geocells hierarchically. First within regions (GID_1), then across regions of the same country (GID_0). To avoid oversampling, each geocell was defined to include at least one complete GID_2 (city-level) administrative unit, even in dense areas like Paris. These geocells were then used to balance the YFCC100M and Landmarks dataset by selecting 5,000 images per source, with at most five images per geocell. For the Landmarks dataset, we also excluded duplicate images of the same landmark to ensure diversity.

### 3.3 PROBING

To examine whether the evaluated models encode geospatial information, we perform linear probing (Alain & Bengio, 2017), a standard mechanistic interpretability technique for Transformer architectures (Gurnee & Tegmark, 2024; Kim et al., 2025). Transformers (Vaswani et al., 2017) consist of sequential blocks that iteratively refine token representations within the residual stream $x^{(l)} \in \mathbb{R}^{t \times d_{\text{model}}}$, where $d_{\text{model}}$ is the hidden dimension of each evaluated model, as an input with $t$ tokens is propagated through the $l$-th Transformer block (Elhage et al., 2021). This refinement is achieved through successive multi-head attention (MHA) and multi-layer perceptron (MLP) layers with residual connections. These layers are often paired with normalization, of which the formulation is omitted for brevity:

$$h_{\text{attn}}^{(l)} = x^{(l)} + \text{MHA}\left(x^{(l)}\right) \tag{1}$$

$$h_{\text{mlp}}^{(l)} = \text{MLP}\left(h_{\text{attn}}^{(l)}\right) \tag{2}$$

$$x^{(l+1)} = h_{\text{attn}}^{(l)} + h_{\text{mlp}}^{(l)} \tag{3}$$

Although $t$ varies across vision models, downstream tasks typically rely on a single summary representation. Usually, the [CLS] token is used in vision models and the final token representation in VLMs. We fit ridge regression models to predict latitude and longitude (in degrees) from layer-wise token summary representations. We report the models' predictive performance using the coefficient of determination ($R^2$). Formally, given hidden representations $\boldsymbol{A}^{(l)} \in \mathbb{R}^{n \times d_{\text{model}}}$ from layer $l$, number of samples $n$, and hidden dimension $d_{\text{model}}$, the regression weights $\mathbf{W} \in \mathbb{R}^{d_{\text{model}} \times 2}$, corresponding to the two-dimensional targets (latitude and longitude), are estimated as:

$$\hat{\mathbf{W}} = \arg\min_{\mathbf{W}} \left\| \boldsymbol{Y} - \boldsymbol{A}^{(l)} \mathbf{W} \right\|_F^2 + \lambda \|\mathbf{W}\|_F^2. \tag{4}$$

Here, $\|\cdot\|_F$ denotes the Frobenius norm, and $\lambda > 0$ is the regularization hyperparameter controlling the strength of the $L_2$ penalty on $\mathbf{W}$. In all of our experiments, $\lambda$ is chosen for each probe using Leave-One-Out cross-validation (Golub et al., 1979).

## 4 EXPERIMENTS

### 4.1 PERFORMANCE ON GEOLOCATION INFORMATION PREDICTION

We perform layer-wise probes on all the evaluated models and report the geolocation prediction performance in Figure 2. We find that models trained jointly on text and images exhibit measurable geospatial representations: the $R^2$ values reach up to 0.8 for *Landmarks* and *streets* (cluster 28), while the average $R^2$ is above 0.4 for the larger models, suggesting some degree of geospatial representations across image types. The average $R^2$ for vision-only models, on the other hand, is mainly below 0.3. Among vision-only models, performance improves with model size, with DINOv2-giant (1B) and the WebSSL-DINO 7B model achieving the best result. This observation suggests

---

[2]https://huggingface.co/datasets/visheratin/google_landmarks_places

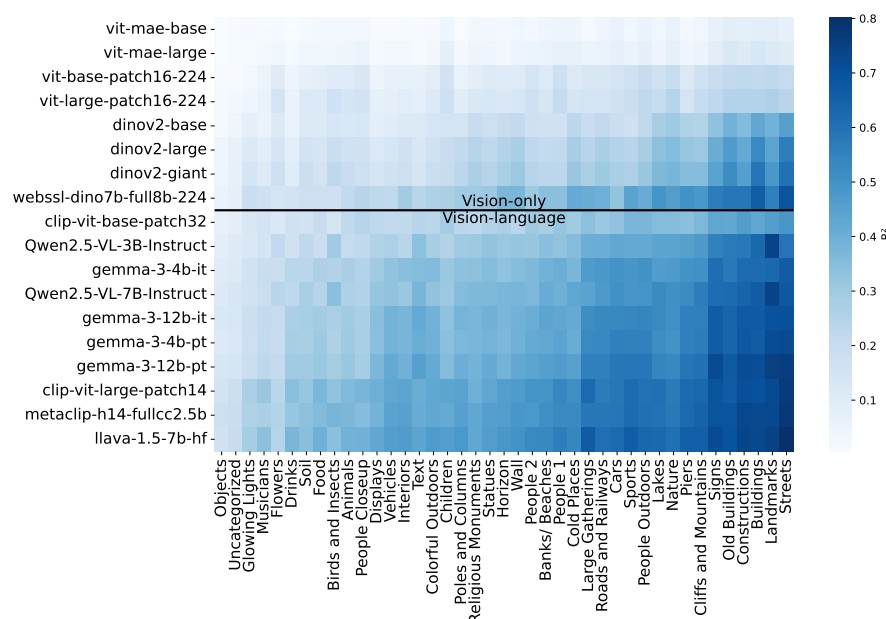

Figure 2: Model performance measured by coefficient of determination $R^2$ across all models. The x-axis shows image clusters based on the YFCC100M dataset and the Google Landmarks dataset, and y-axis lists the models compared. Higher $R^2$ values (darker colors in the heatmap) indicate better geolocation-prediction accuracy across clusters.

that, when trained on a scale, geospatial representation can be learned from images alone. When compared with VLMs, DINOv2-giant is outperformed even by the much smaller CLIP-base model, suggesting the effectiveness of language pretraining in implicitly learning geospatial representations.

The cluster-wise performance presented in Figure 2 indicates that the relative difficulty of each cluster in YFCC100M remains consistent across various model architectures. For instance, the *streets* cluster, *building* cluster, and the *Landmarks* dataset consistently show higher $R^2$ across models, while the *objects* cluster contains little information that can serve as clues for geo-localization. Interestingly, the *signs* and *text* clusters show a degree of localizability for VLMs not observed for vision-only models.

Figure 3 shows image samples positioned according to their $R^2$ values for the largest model of each model family. Highly localizable images tend to be famous landmarks, e.g., pyramids, and open spaces with pieces of architecture or nature. Meanwhile, close-ups of objects and food images are the least localizable. Notice that for CLIP, the cluster of figures containing signs achieves a notable degree of localizability.

### 4.2 GEOSPATIAL REPRESENTATIONS ACROSS LAYERS

To investigate where geospatial representation emerges within model architectures, we analyze probe performance across layers for both vision-only and multimodal models. In vision models such as ViT and DINOv2, geospatial representations tend to develop progressively with increasing layer depth, as evidenced by a consistent rise in probe $R^2$ values across all settings (see Figure 4(a)). For VLMs, however, an interesting observation emerges: the $R^2$ values increase only up to a certain point before stagnating, and in the case of Gemma, the $R^2$ values decrease throughout the later layers, regardless of image characteristics. This likely reflects the model's tendency to deprioritize geographic signals in the absence of a textual prompt, thus neglecting spatial information not essential for text generation. A similar, though less pronounced, effect is observed in LLaVA, where geolocation signals diminish as the image transitions into the language modeling component.

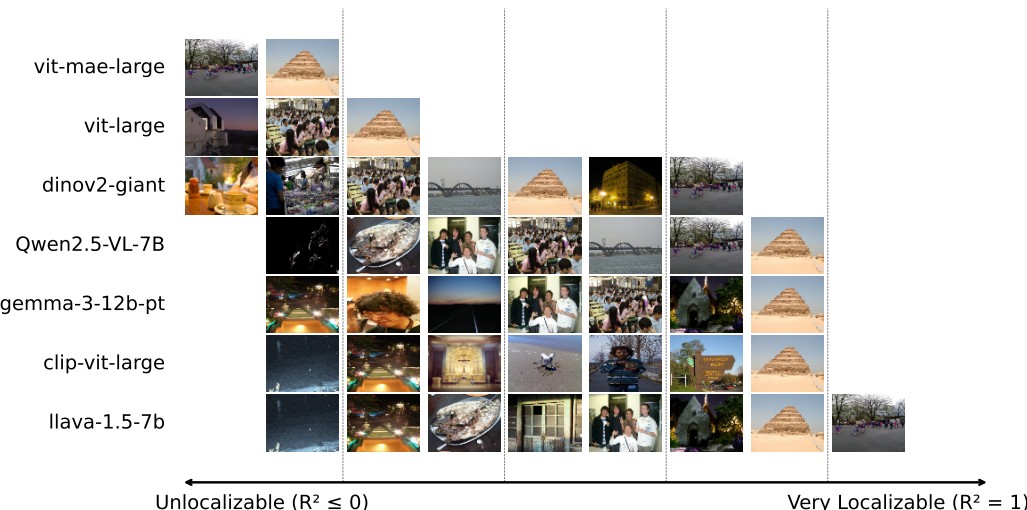

Figure 3: Illustration of image cluster localizability for each model, where each image serves as a representative of a cluster (Appendix B). The images to the right represent clusters with better localizability (higher $R^2$). For high-performing models, highly localizable landmarks usually achieve the highest performance.

## 4.3 STEERING THE MODELS VIA TEXT PROMPTS

In prior experiments, we observed that the geospatial information in the model residuals, particularly VLMs, degrades over layers without a textual prompt. This leads to, in some extreme cases, for example, Gemma, a negative $R^2$, suggesting that there is no linear mapping from the model residuals to geolocation coordinates.

To check if this effect occurs because of the absence of a textual prompt related to geospatial information, we prompt the models with the query "Guess the latitude and longitude of this image. Answer only with the coordinate tuple (lat, long)". Figure 4(b) shows the per-layer $R^2$ of VLMs when prompted this way. We observe that, in this setting, $R^2$ does not decrease as drastically for Gemma and LLaVA. In fact, for LLaVA, it starts increasing over the textual layers. Interestingly, for Qwen the performance drastically increases, leading to $R^2$ as high as 0.88 for the *Landmarks* dataset. This effect suggests that textual and image representations of geospatial representation might be entangled in the model's latent space, especially when activated using textual prompts related to geospatial tasks. We expand on this discussion in Appendix G.

## 4.4 ISOLATING GEOSPATIAL-SPECIFIC COMPONENTS FROM EMBEDDINGS

The linear probes operate on high-dimensional representations. In our experiments, the five selected models have $d_{\text{model}}$ of 768 for CLIP-ViT-large, 1,024 for LLaVA-1.5, 2,048 for Qwen2.5-VL-3B, 3,584 for Qwen2.5-VL-7B, and 1,536 for DINOv2-giant. To explore how latent space contributes to geospatial information, we fit ridge regression probes using only a proportion of the original dimensions $p \in \{0.1, 0.2, \ldots, 1.0\}$, selecting the top $p$ dimensions ranked by the absolute coefficients of the trained probe.

We report the predictive performance in terms of $R^2$ in Figure 5 as a function of the retained feature proportion $p$, showing that $R^2$ increases with $p$ and saturates well before using the entire feature set. Across

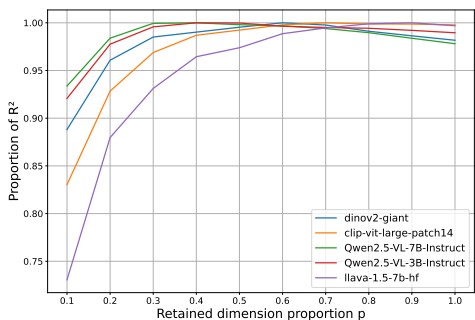

Figure 5: Probe predictive performance $R^2$ as a function of the retained feature proportion $p$, illustrating the capacity of the embedding subspace needed to reach the maximum $R^2$ for both vision-only and VLMs. Higher values of $p$ correspond to a larger subset of the latent representation.

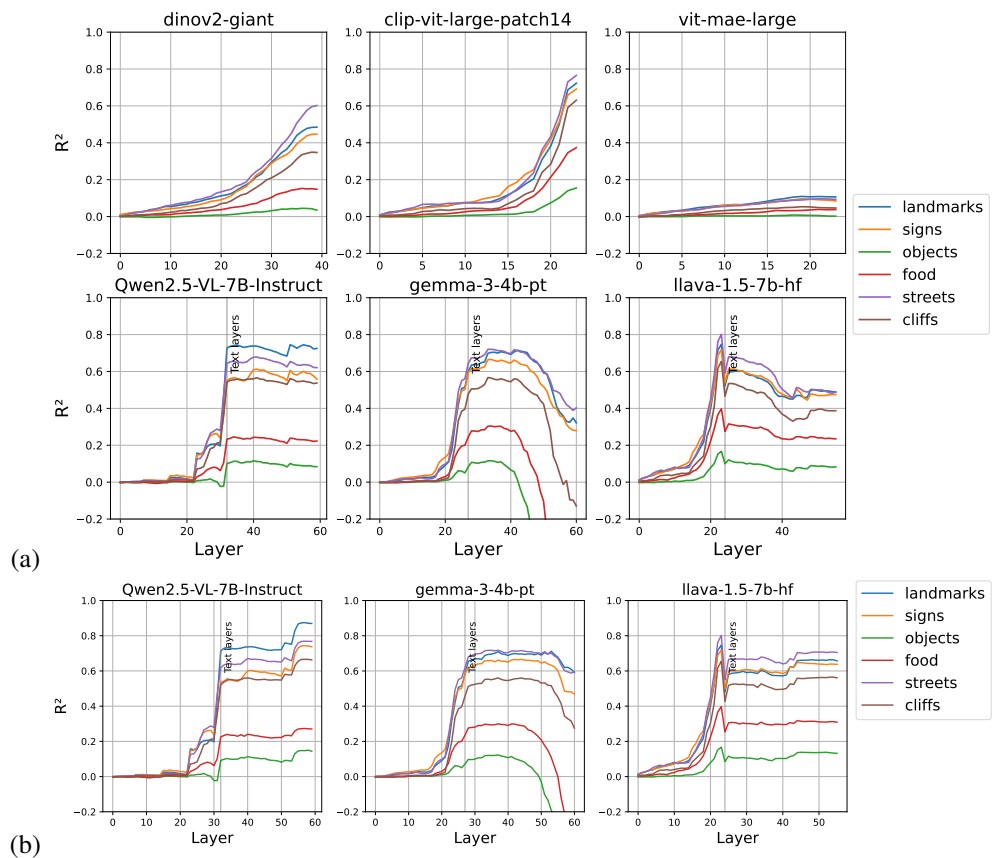

(a)

(b)

Figure 4: Probe $R^2$ performance by layer of the models for different clusters and datasets with varying levels of localizability. (a) $R^2$ performance when no textual prompt is given. (b) $R^2$ performance when adding a textual prompt to the input asking the model to predict the image geolocation. The $R^2$ is kept stable throughout the last layers when compared to the decaying performance observed in the non-prompting setup.

all models, we find that $p \approx 0.4$ (about 40% of dimensions) are sufficient to recover nearly the maximum $R^2$, indicating that geospatial information is concentrated in a compact subset of dimensions rather than uniformly distributed throughout the embeddings. For Qwen2.5-VL variants, specifically, 90% of their best predictive performance is still observed using only the top 10% of the features.

## 4.5 STEERING THE MODEL GENERATION THROUGH REPRESENTATION SWAPPING

We examine the possibility of steering the text generated by a multimodal foundation model by swapping the top $p$ feature dimensions related to geospatial reasoning. In this case study, the geospatial location in the predicted text should be changed, leaving other semantic information unchanged. We use Qwen2.5-VL-3B, an open-weight VLM that supports a targeted intervention in its *residual stream* or the additive hidden state passed through the transformer layers. As illustrated in Figure 6, given a *source* image and a *target* image, we replace the top geospatially relevant feature dimensions of the *source* residual-stream summary with the corresponding coordinates from the *target*, and evaluate the resulting changes in the generated text.

Let $t^\star$ be the last non-padding input token (the summary token for Qwen2.5-VL-3B), and let $\boldsymbol{A}^{(1)}_{\text{source}, t^\star}, \boldsymbol{A}^{(1)}_{\text{target}, t^\star} \in \mathbb{R}^{d_{\text{model}}}$ be the layer-1 residual-stream vectors for source and target images, respectively. Given $g \subseteq \{1, \ldots, d_{\text{model}}\}$ as the index set of geospatially informative dimensions (with proportion $p = |g|/d_{\text{model}}$) and its complement $g^c = \{1, \ldots, d_{\text{model}}\} \setminus g$, we intervene by replacing the source residual with the target residual on the dimensions in $g$ as follows:

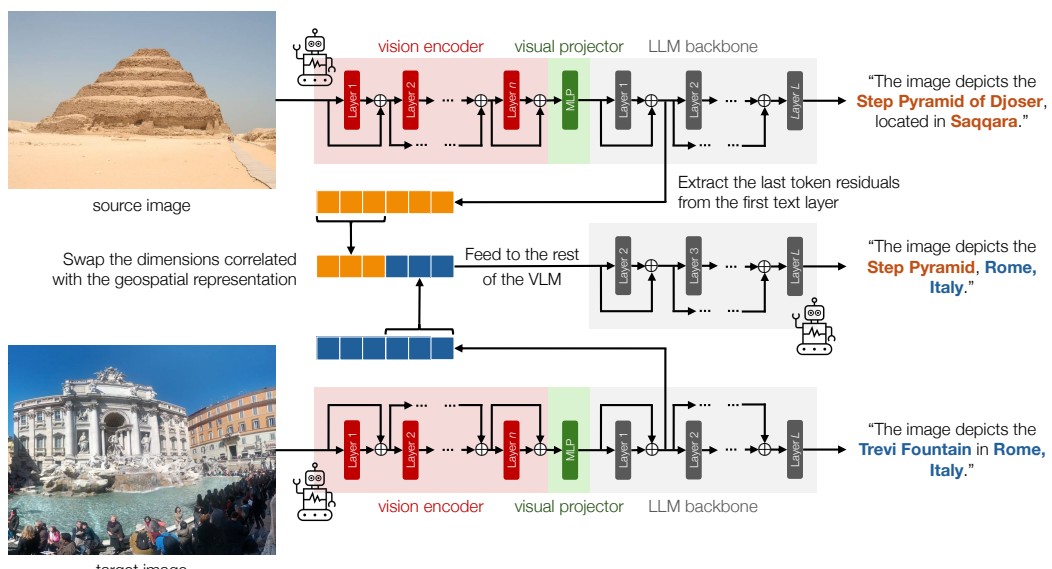

Figure 6: Schematic illustration showing that editing the geospatial representations (through dimension swapping) changes the perceived geolocation during token generation of VLMs. We demonstrated this finding on Qwen2.5-VL-3B with the methodology described in Section 4.5.

$$\tilde{\boldsymbol{A}}^{(1)}_{\text{source},t^\star} = \boldsymbol{A}^{(1)}_{\text{source},t^\star} \odot \mathbb{1}_{g^C} + \boldsymbol{A}^{(1)}_{\text{target},t^\star} \odot \mathbb{1}_g, \tag{5}$$

where $\odot$ denotes the element-wise Hadamard product and $\mathbb{1}_g$ an indicator vector with entries 1 for indices in $g$ and 0 elsewhere. For brevity, we omit explicit indexing of the selected dimensions $g$. We then continue the forward pass from layer $2, \ldots, L$ using $\tilde{\boldsymbol{A}}^{(1)}_{\text{source}}$ to decode the output text.

We show an example in which we can successfully steer the model in Figure 6. By swapping 50% of geospatial representations from an image of the Step Pyramid of Djoser with the geospatial representation from an image of the Trevi Fountain, the model generates the following output text: "The image depicts the Step Pyramid, Rome, Italy", altering the location of the pyramid from Saqqara to Rome. Even though our experiments show the possibility of successfully steering the model, we observed that as the text generated becomes longer, the generation becomes unstable. In some cases, the model starts to generate repetitive text or descriptions that mix the source and target locations. These results open future avenues for investigating how geospatial representations are coupled with representations related to other types of information during text generation. We discuss more details in Appendix H.

## 4.6 Downstream Task Performance

Finally, we inspect how the quality of geospatial representations may influence downstream task performance, as these models are usually fine-tuned for specific downstream applications. For this analysis, we investigate a task that requires geospatial awareness: country identification.

Using the landmarks dataset, we extract country information for each picture and then subsample the dataset so that at most 100 pictures are selected for each country. Then, we finetune one large model from each studied vision-only family (ViT-MAE, ViT, and DINOv2) in addition to CLIP-ViT-large and DINOv2-giant (for the full details, see Appendix I). The models are chosen such that all take the same inputs and have similar model size, making the results comparable. We report the results of each model in Table 1. We observe that the performance of the models follows the order of $R^2$ obtained for our probe analysis, with ViT-MAE having the worst performance, while CLIP has the best performance. This corroborates the hypothesis that the presence of geospatial representations in the models is desirable for their use in downstream tasks.

## 5 DISCUSSION

Our findings demonstrate that the training methodology is crucial for learning geospatial representations in vision-only models and VLMs, as demonstrated in Figure 2. Models that incorporate textual supervision consistently achieve the best performance. Vision-only models improve with scale; larger models like DINOv2-giant outperform both DINOv2-large and DINOv2-base. In contrast, VLMs do not exhibit a clear correlation between model

Table 1: Fine-tuning performance for the country identification task.

| Model | Test Acc. | Val. Loss | Train Loss |
|---|---|---|---|
| ViT-MAE-large | 0.15 | 3.35 | 2.344 |
| ViT-large | 0.23 | 3.17 | 1.346 |
| DINOv2-large | 0.29 | 2.55 | 0.009 |
| DINOv2-giant | 0.32 | 2.78 | 0.001 |
| CLIP-large | 0.36 | 2.39 | 0.009 |

scale and probing performance. This suggests that supervision signals, particularly language, are a primary factor in learning strong geospatial representations in these models. In this way, relating our findings to the Platonic Representation Hypothesis Huh et al. (2024), even though scaling vision-only models improves performance, textual supervision enhances the efficiency for learning geospatial representations. | APe2.Q1

The optimal layer for extracting geospatial representations depends on the model family and the presence of a textual prompt. In many geolocation applications, the input image is given without a textual prompt (Figure 4(a)). For these cases, vision-only models perform best when using representations from their deepest layers. However, for VLMs, choosing which geospatial representation to use is not clear and varies between models. Across models, the layers immediately after the textual stream consistently achieve good performance. With a textual prompt, the performance in VLMs remains more stable across the post-textual layers (Figure 4(b)).

The results of our analysis have direct implications for model selection and methodology in a range of geospatial applications. For current pipelines that use traditional vision-only models and small datasets, leveraging representations from VLMs can significantly improve performance. As these representations are implicitly learned from vast datasets, they serve as strong representations for sample-efficient pipelines involving fine-tuning cases where labeled data are scarce. Moreover, our work highlights multimodal learning as a critical direction to build world models, which could be used to improve our understanding of complex social problems and empower new technologies.

Finally, the growing capability of these models poses significant privacy risks and fairness implications. These models have spatial imbalance on geolocation performance, with lower performance for underrepresented regions (Appendix C). Additionally, malicious actors could exploit these models to extract precise location data from images, enabling stalking and threats against individuals. The potential for mass surveillance is also a serious concern, where governments and corporations could likewise track individuals' locations and behaviors through their photos. These ethical risks underscore the need for robust regulatory policies that mandate transparency in model use and enforce explicit user consent to ensure safe deployment. | APe2.W2

## 6 CONCLUSION

This study demonstrated that textual supervision significantly enhances geospatial representations in vision-language models. Through a systematic analysis of vision-only architectures, VLMs, and large-scale multimodal foundation models, we studied how geospatial understanding emerges across model families. Through layer-wise probing, we revealed that multimodal models consistently exhibit high performance for images that are localizable. Furthermore, our analysis indicated that a small subset of hidden dimensions is responsible for encoding critical geospatial features, suggesting a potential pathway for model steering and editing. In summary, our work demonstrated that multimodal learning plays an important role in improving geospatial AI. However, using these models in real-world settings should include safeguards to protect privacy and ensure fairness. As applications involving location-aware image understanding—such as environmental monitoring, urban planning, and disaster response—continue to grow, this use case is expected to become increasingly important. Future research could explore how these models handle other types of images, such as satellite data.

## REPRODUCIBILITY STATEMENT

The code for reproducing our results is available through an anonymous repository for validation[3], and all datasets used in this study are publicly accessible.

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

## A    MODEL DETAILS

Details about the models evaluated in this work are given in Table 2.

Table 2: Models evaluated in this work—ViT (Dosovitskiy et al., 2021), ViT Masked Autoencoder (He et al., 2022), DINOv2 (Oquab et al., 2023), CLIP (Radford et al., 2021), LLaVA-1.5 (Liu et al., 2024), Qwen2.5 (Bai et al., 2025), and Gemma 3 (Kamath et al., 2025)—spanning vision-only and vision-language modalities with different training paradigms. Each family is evaluated with at least two size variants to examine the effect of model scale on learned representations.

| Model | Modality | Training Methodology |
|---|---|---|
| ViT | Vision-Only | Supervised pretraining on ImageNet-21k followed by fine-tuning on ImageNet-1k. |
| ViT Masked Autoencoder | Vision-Only | Self-supervised training using masked autoencoding. |
| DINOv2 | Vision-Only | Self-supervised learning using a teacher-student framework. |
| CLIP | Vision-Language | Trained on image-text pairs using contrastive learning. |
| LLaVA-1.5 | Vision-Language | Combines a CLIP-based vision encoder with a language model via vision-language alignment, followed by instruction tuning. |
| Qwen2.5 | Vision-Language | CLIP-based pretraining enhanced with vision-language alignment and end-to-end instruction tuning. |
| Gemma 3 | Vision-Language | Uses SIGLIP-based vision-text pretraining followed by alignment with instruction-tuned language models. |

To extract the inner representations of the model, for the probing experiments, we use a single image as input and no prompt in the VLMs; then we conduct a single forward pass over the entire model, extracting all residuals $A^{(l)}$ for the last token for both the vision and the text components when applicable. A similar setup is adopted for the prompting experiments, but with added prompt tokens. Since we are not interested in generating text, we use no specific parameters (e.g., temperature) for the VLMs for the probing experiments. For the generation experiments, we use a very low temperature (0.0001) to reduce randomness.

All experiments were run on a single Nvidia A100 GPU using the HuggingFace implementation of each model. The models that only take images as input were run at full precision, while the VLMs (Gemma, Qwen, and LLaVA) were run in bfloat16 precision.

## B    CLUSTERING DETAILS

Given the massive size of the YFCC100M dataset, its content was divided into semantic clusters for a better and more comprehensive analysis. For this, we started by extracting embeddings from the

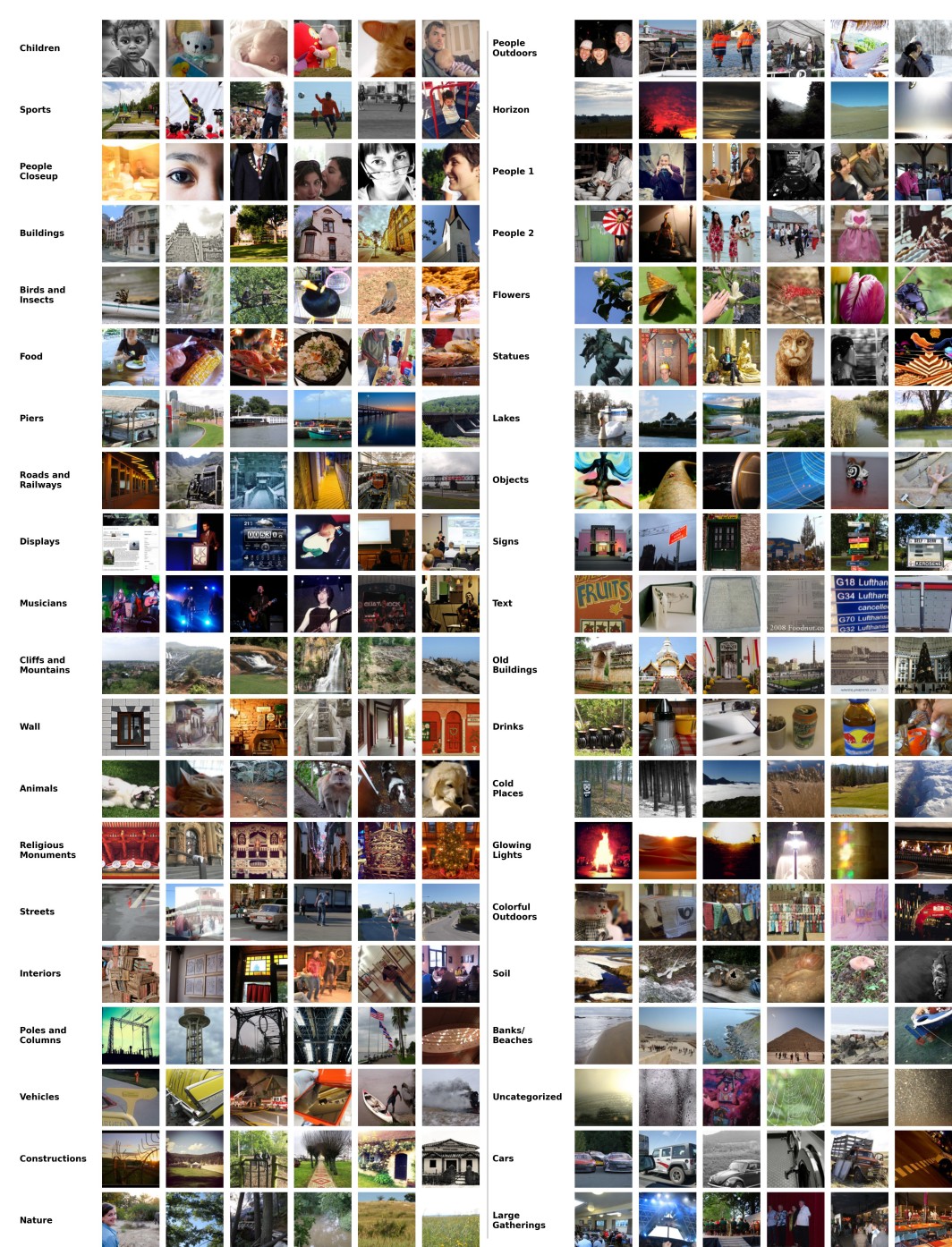

Figure 7: Random samples for each of the 40 clusters obtained for the YFCC100M dataset.

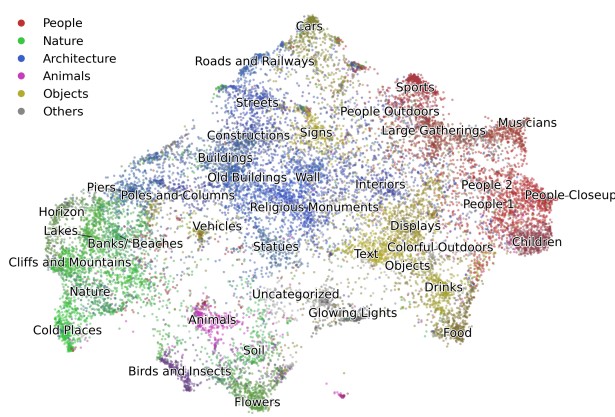

Figure 8: Two-dimensional visualization of the 40 clusters obtained using UMAP and colored according to macro-category.

final convolutional layer of a ResNet-152 pretrained on ImageNet. Then, the resulting embeddings were $L_2$-normalized, reduced to 100 dimensions using PCA, and clustered with a standard $k$-means algorithm. To choose the optimal $k$, we tested 19 values ($k = 10, 15, \ldots, 100$), and following (Thorndike, 1953), computed the within-cluster sum of squares (WCSS) as follows:

$$\text{WCSS}(k) = \sum_i D_i = \sum_{i=1}^{N} \min_{c \in \{1, \ldots, k\}} \|\mathbf{x}_i - \boldsymbol{\mu}_c\|^2,$$

which corresponds to the sum of squared distances for each point to its nearest cluster center. The value of $\text{WCSS}(k)$ decreases monotonically as $k$ increases, meaning that the optimal $k$ is not the one which minimizes $\text{WCSS}(k)$, but rather the one at which the decrease plateaus. This point corresponds to the "elbow" of the curve, which in our case was $k = 40$.

A complete overview of the resulting clusters can be seen in Figure 7. In total, eight clusters represented people, including separate clusters for sports, musicians, and children, as well as another cluster for large gatherings. Another 11 clusters were related to man-made structures and architecture, including different types of buildings, walls, streets, and decorations. Animals were represented in two groups: one for birds and insects, and the other for larger animals. Eight clusters depicted different aspects of nature and different landscapes, such as mountains, lakes, beaches, soil, etc. Finally, seven clusters showed different types of objects, mainly food, drinks, displays, and vehicles. The remaining three clusters did not appear to exhibit any clear semantic relationship.

For better visualization of the clusters obtained, we projected a small subset of 500 points per cluster into a 2D space using Uniform Manifold Approximation and Projection (UMAP) (McInnes et al., 2018). The generated plot (Figure 8) uses different shades of the same base color to represent clusters belonging to the same macro-category (people, nature, architecture, objects, animals, and others). Clusters associated with nature, architecture, and people are tightly grouped and occupy a distinct region of the latent space, further indicating that our clustering is semantically coherent. The majority of object clusters also lie in a clearly defined area, though there is some variability depending on the image background. Notably, cars and signs—traditionally found in urban settings—appear close to architecture clusters, while general vehicles, e.g., trains and airplanes, appear in between nature and architecture. The two animal clusters are in a subregion of nature, which is consistent with their broader photographic context, as these images are typically taken in green areas. One exception is the "People Outdoors" clusters, which, though grouped together with other people clusters, extend towards both architecture and nature regions, depending on the broader context of the images.

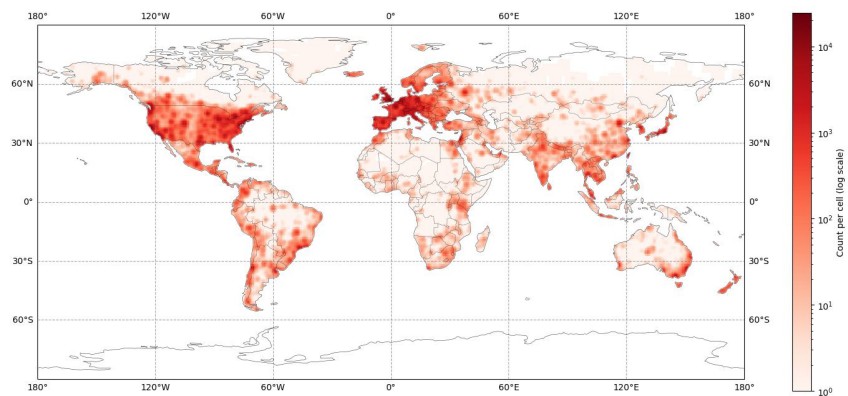

Figure 9: Spatial distribution of all samples from our subset of YFCC100M.

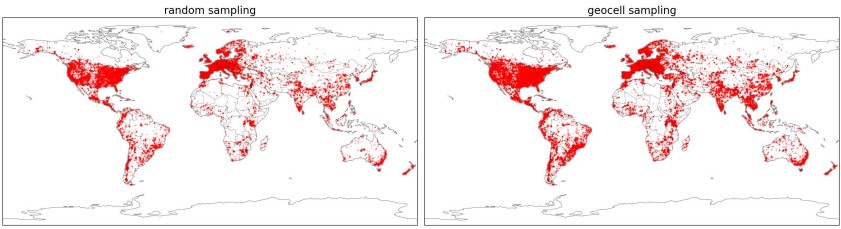

Figure 10: World coverage of random sampling (left) and geocells-based sampling (right).

## C    DATASET DESCRIPTION

### C.1    YFCC100M

The YFCC100M dataset comprises approximately 99.2M images published on Flickr between 2004 and 2014 under a Creative Commons license (both commercial and non-commercial). It is a highly diverse set of photographs that depict natural and urban environments, people, objects, and everyday events, taken by a mix of professional photographers and casual users. In our experiments, we used a subset of 4,233,900 images, which preserves the diversity of the original set while offering reliable geolocation annotations.

The spatial distribution of the data set is highly unbalanced (Figure 9), with the majority of the samples concentrated in a few key regions. Together, the G7 countries account for more than 57% of all samples, a figure that rises to 78% with the addition of the rest of the European countries. Asia is the third most represented continent with close to 500k images (12.1% of the data), more than half of which are from East Asia. In contrast, Central Asia and the Middle East are particularly underrepresented, with no country in either region contributing more than 15k samples.

South America accounts for just over 180k images (4.2% of the data), with a sample distribution that closely matches that of the continent's population. The main outliers are Chile, overrepresented in 17% of the data versus 4.5% of the population, and Venezuela, underrepresented at less than 3% of the samples despite being 6.5% of the total population. Africa contributes approximately 68k images (1.6% of the dataset), with only 12 countries represented by more than 1k samples. Finally, Oceania provides 136k (3.2% of the data) samples, almost entirely from Australia and New Zealand, which together account for 131k. We note that, although there is some variation, all clusters exhibit roughly the same imbalance. After sampling, the balance is marginally improved. Of the 200k total sampled images (5k per cluster), 37k (18.5%) are from Asia, almost 14k (7.0%) are from South America, 8.9k (4.5%) are from Oceania, and 7.4k (3.7%) are from Africa, while the participation of G7 countries decreases from 57% to 44%. Figure 10 shows the effects of these improvements, with our sampling methodology leading to better world coverage than a purely random strategy.

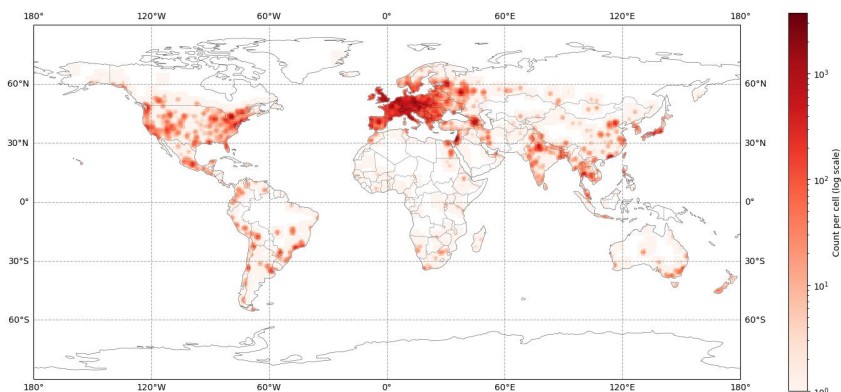

Figure 11: Spatial distribution of all samples from our subset of Google Landmarks V2.

## C.2 GOOGLE LANDMARKS

Google Landmarks V2 contains over 5M images of over 200k landmarks across the globe, collected mostly from Wikimedia Commons[4]. As the coordinates of images are not typically available on Google Landmarks, we used a subset of 581,215 geolocalized images from 15,453 different labeled landmarks. Though photos of relevant sites are expected to be localizable, the dataset also contains some non-localizable images, such as close-up shots of animals in a national park, as well as paintings and statues.

This data set has a pronounced spatial imbalance, as shown in Figure 11. Samples are mainly concentrated in Europe, which alone accounts for over 67% of our subset, followed by Asia (15%), North America (11%), South America (2.5%), Africa (1.1%), and Oceania (1.0%). Outside of Europe, samples are disproportionately concentrated in a few major cities, with less populated areas remaining mostly uncovered. For our experiments, we selected a single random image from each landmark and proceeded to sample 5k images as outlined in Section 3.2.1. The sampling procedure preserved the same continent-level spatial bias present in the original dataset, though the concentration of samples in major cities was reduced.

To investigate the effects of spatial imbalance on geolocation performance, we compute several spatial sensitivity metrics in our subset of Google Landmarks V2. We focus on three complementary indicators: (i) marked spatial self-information (SSI) geo-bias score (Wu et al., 2024), which measures the extent to which a model's incorrect predictions cluster spatially beyond what would be expected from random errors; (ii) median geodesic error (in kilometers); and (iii) median proximity error (Gurnee & Tegmark, 2024), which quantifies the fraction of candidate predictions that lie closer to the ground-truth location than the model's own prediction (i.e., a rank-based error). All metrics were computed at the sample level and aggregated both per continent and over the full dataset. APe2.W2

For SSI geo-bias, we used the original implementation, made available by the authors [5], and defined a prediction as *good* if its geodesic error was smaller than 1,000 km. A value of 0 indicates that prediction errors are spatially random, while higher values reflect increasingly clustered errors. The resulting scores (visually illustrated in Figure 12) show that most vision-only models are heavily spatially biased, as their predictions tend to be clustered around the densest parts of the dataset. A more in-depth look at our best-performing VLMs is provided in Table 3. Overall, SSI values tend to be smaller for samples originally located in Europe, suggesting that errors in these samples behave more randomly compared to those in other regions. However, even within Europe, values remain relatively high, indicating that the model performs better in specific subregions or that they disproportionately succeed in urban areas, where sample concentration is higher.

Though SSI is a good indication of how non-random spatial distribution of prediction errors is, it does not reflect the magnitude of those errors across regions. To assess this, we examine geodesic and proximity errors of our selected VLMs, reported in Tables 4 and 5, respectively. Across all

---

[4]https://commons.wikimedia.org/
[5]https://github.com/seai-lab/GeoBS

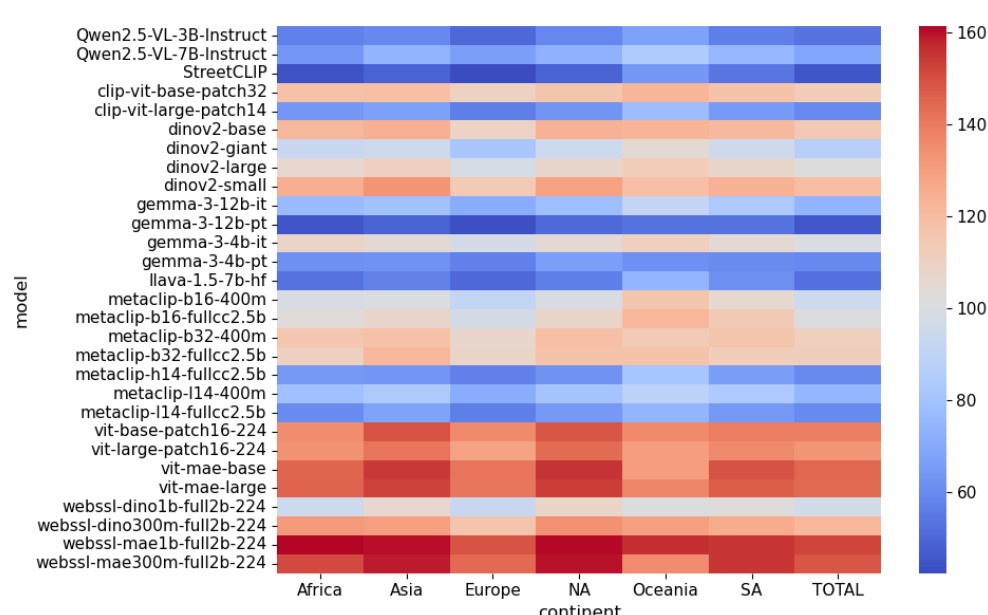

Figure 12: Marked SSI geo-bias values for all models on our subset of Google Landmarks V2

Table 3: Marked SSI geo-bias score per continent for a representative subset of VLMs with a $95\%$ confidence interval (lower is better). NA and SA indicate North and South America, respectively.

| Model | Africa | Asia | Europe | NA | SA | Oceania |
|---|---|---|---|---|---|---|
| Qwen2.5-VL-3B | $56.7 \pm 5.82$ | $59.3 \pm 2.05$ | $50.1 \pm 0.90$ | $58.9 \pm 1.97$ | $56.4 \pm 4.19$ | $67.2 \pm 8.16$ |
| Qwen2.5-VL-7B | $63.9 \pm 4.79$ | $73.7 \pm 2.35$ | $66.2 \pm 0.91$ | $73.1 \pm 2.06$ | $74.9 \pm 4.57$ | $84.0 \pm 6.44$ |
| LLaVA-1.5-7B-HF | $52.6 \pm 5.02$ | $57.3 \pm 2.22$ | $49.9 \pm 0.88$ | $56.5 \pm 2.04$ | $61.1 \pm 4.26$ | $73.9 \pm 9.86$ |
| CLIP-ViT-large | $64.0 \pm 5.18$ | $66.8 \pm 2.07$ | $56.4 \pm 0.89$ | $63.1 \pm 1.99$ | $77.4 \pm 6.77$ | $64.8 \pm 4.19$ |
| Gemma-3-12B-PT | $44.7 \pm 4.49$ | $48.5 \pm 1.74$ | $43.2 \pm 0.77$ | $50.2 \pm 1.67$ | $53.4 \pm 4.13$ | $53.3 \pm 6.68$ |

models, geodesic error was consistently and significantly lower in Europe, further reinforcing that performance tends to be better in this region. In contrast, South America and Oceania were the most challenging continents for all models, indicating that the high concentration of samples in the Northern Hemisphere could have a significant impact on the success rate of Southern Hemisphere predictions.

As expected, proximity error is the highest in Europe, as the higher density makes retrieval ranking more difficult. On the other hand, South America and Africa had surprisingly low proximity errors, indicating that, though the models may not be good at precisely predicting coordinates in these regions, they generally stay within the correct part of the world. Oceania, by contrast, exhibits much higher proximity errors, indicating that even correctly identifying samples as being in Oceania might be challenging enough for the models.

To better understand how these continent-level trends manifest spatially, we also visualize the median geodesic error of Gemma-3-12B-PT on a $4° \times 4°$ grid (Figure 13, left). As suggested by the aggregated metrics, errors are consistently lower in Europe and noticeably higher throughout the Southern Hemisphere. North America and Asia exhibit less uniform behavior, characterized by a mix of high- and low-error regions. Figure 13 (right) shows the model's predicted coordinates, colored by their ground-truth continent. This plot corroborates our earlier observations that, except for Oceania, the model typically places predictions within the correct broad region of the world despite large geodesic errors in certain areas.

Table 4: Median geodesic error (km) per continent for a representative subset of VLMs. Lower is better.

| Model | Africa | Asia | Europe | N. America | S. America | Oceania | Total |
|---|---|---|---|---|---|---|---|
| Qwen2.5-VL-3B | 2185.6 | 1686.2 | 800.1 | 1966.5 | 4119.1 | 9212.8 | 1029.1 |
| Qwen2.5-VL-7B | 2246.5 | 1688.7 | 959.9 | 1890.7 | 4225.0 | 9720.4 | 1173.5 |
| LLaVA-1.5-7B-HF | 2485.8 | 1754.8 | 799.4 | 1785.1 | 5032.5 | 7140.4 | 1030.4 |
| CLIP-ViT-large | 2440.6 | 1950.1 | 827.9 | 2000.7 | 8151.6 | 5257.9 | 1091.9 |
| Gemma-3-12B-PT | 1865.9 | 1546.3 | 752.9 | 1768.0 | 3965.3 | 7259.0 | 962.6 |

Table 5: Median proximity error per continent for a representative subset of VLMs. Lower is better.

| Model | Africa | Asia | Europe | N. America | S. America | Oceania | Total |
|---|---|---|---|---|---|---|---|
| Qwen2.5-VL-3B | 0.014 | 0.042 | 0.224 | 0.079 | 0.026 | 0.120 | 0.119 |
| Qwen2.5-VL-7B | 0.016 | 0.043 | 0.309 | 0.080 | 0.026 | 0.113 | 0.144 |
| LLaVA-1.5-7B-HF | 0.011 | 0.043 | 0.223 | 0.073 | 0.027 | 0.031 | 0.113 |
| CLIP-ViT-large | 0.015 | 0.046 | 0.242 | 0.079 | 0.054 | 0.028 | 0.129 |
| Gemma-3-12B-PT | 0.012 | 0.037 | 0.201 | 0.072 | 0.025 | 0.039 | 0.104 |

## D  EFFECT OF ARCHITECTURE AND SCALE

In our experiments, we employ a variety of models with different training objectives, architectures, scales, and training data. By selecting a large array of models, we find compelling evidence that the use of textual data might be one driving factor for improved geospatial representations inside each model. For instance, even models with more than 1B parameters (DINOv2-giant), when trained solely on visual data in a self-supervised manner, fail to outperform much smaller models, such as CLIP-base, for most of the subsets of our data.

Still, it is not clear how much of this can be attributed to the images used to train each model, rather than the text. For this reason, in this section, we make a more strict comparison, using models with very similar architectures and scale, pretrained on the same dataset, with the major difference being the training objective and the inclusion (or not) of text. We select the MetaCLIP and Web-SSL DINO and MAE models, which are all trained on the MetaCLIP dataset, a collection of image-caption pairs sampled from Common Crawl. The summary of probing results for different model configurations is presented in Table 6. We also show the architectural details for each model in Table 6 for easier comparison.

The results for this analysis match our previous observations, with the MAE model, on average, showing the weakest evidence of internal geospatial representations, followed by DINO and finally by CLIP. Among vision-only objectives, we note that the DINOv2 trained model is once again associated with increased quality of geospatial representations when compared to the MAE, suggesting that textual supervision, albeit more effective, is not the only way to introduce this kind of infor-

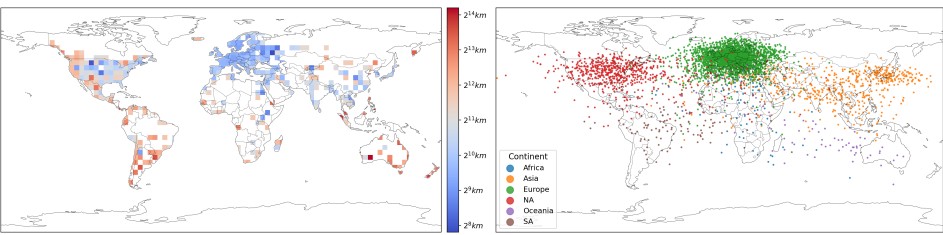

Figure 13: More in-depth look into Gemma-3-12B-PT predictions in our subset of Google Landmarks V2. Median geodesic error per $4° \times 4°$ grid (left). All predictions are colored by the ground truth continent (right).

Table 6: Probe $R^2$ in the landmarks dataset and architectural/training details for models trained on MetaCLIP data. All models have a $224 \times 224$ image resolution.

| Architecture | Objective | Parameters | Patch Size | Train Steps | $R^2$ |
|---|---|---|---|---|---|
| MetaCLIP-base | Contrastive | 87M | 16 | | 0.53 |
| MetaCLIP-large | Image-Text | 300M | 14 | 12.8B | 0.70 |
| MetaCLIP-huge | Pretraining | 600M | 14 | | 0.73 |
| Web-SSL DINO-300M | | 300M | | 2B | 0.26 |
| Web-SSL DINO-1B | DINOv2 Loss | 1B | 14 | 2B | 0.39 |
| Web-SSL DINO-7B | | 7B | | 8B | 0.56 |
| Web-SSL MAE-300M | Masked | 300M | 16 | 2B | 0.11 |
| Web-SSL MAE-1B | Autoencoder | 1B | 14 | | 0.14 |

mation. Moreover, MetaCLIP large and huge models, trained with a smaller compute budget, still outperform the DINO-7B model, trained on the same dataset, suggesting the usefulness of textual supervision. Further developments in self-supervised learning for ViTs may continue to close this gap in the future, especially as scaling up the number of parameters seems to be correlated with increased performance for this task.

## E  LINEARITY OF GEOSPATIAL FEATURES

In our experiments, we focus on the linear probing for geospatial representation in vision-only and vision-language models, finding that the latter group has internal representations that can be mapped to real-world locations, while the former group is much more limited in this regard. However, it could be the case that this happens only because vision-only models represent this kind of information non-linearly. To explore this possibility, we also train non-linear probes (one hidden layer MLP regression) to check whether vision-only models may be representing geospatial features differently.

Figure 14 shows the performance ($R^2$) for the non-linear probes for each model. Using non-linear probes did not result in a significant increase in $R^2$ for any model, thus strengthening the claim that vision-language models' representations encode geospatial information better.

## F  ABLATION STUDIES

Our probing setup utilizes the summary representation of the input image, which is either the [CLS] token or the final token representation. However, it could be the case that a geospatial representation emerges across different tokens corresponding to different image patches. To control for this, we also train the same probes using the concatenation of the min and max pooling across all tokens as the inputs for the ridge regression. Figure 15 shows the results across our datasets. It is possible to see that, when compared to the default probing approach, the $R^2$ is actually smaller for the vision-only models, suggesting that the summary token is adequate as a set of features for the probing setup.

## G  ADDITIONAL PROMPTING RESULTS

In Section 4.3, we show that by using specific textual prompts, such as "Guess the latitude and longitude.", we can strengthen the performance of the probes on the latter text layers of VLMs, suggesting improved representation of geospatial features. In this section, we expand on that by comparing the usage of prompts with varying levels of relevance for coordinate prediction.

Figure 16 shows the $R^2$ for the probes trained on models using five different prompt configurations:

- **None:** No prompt, only the image.
- **Random:** 20 tokens sampled randomly for each picture (consistent across models).

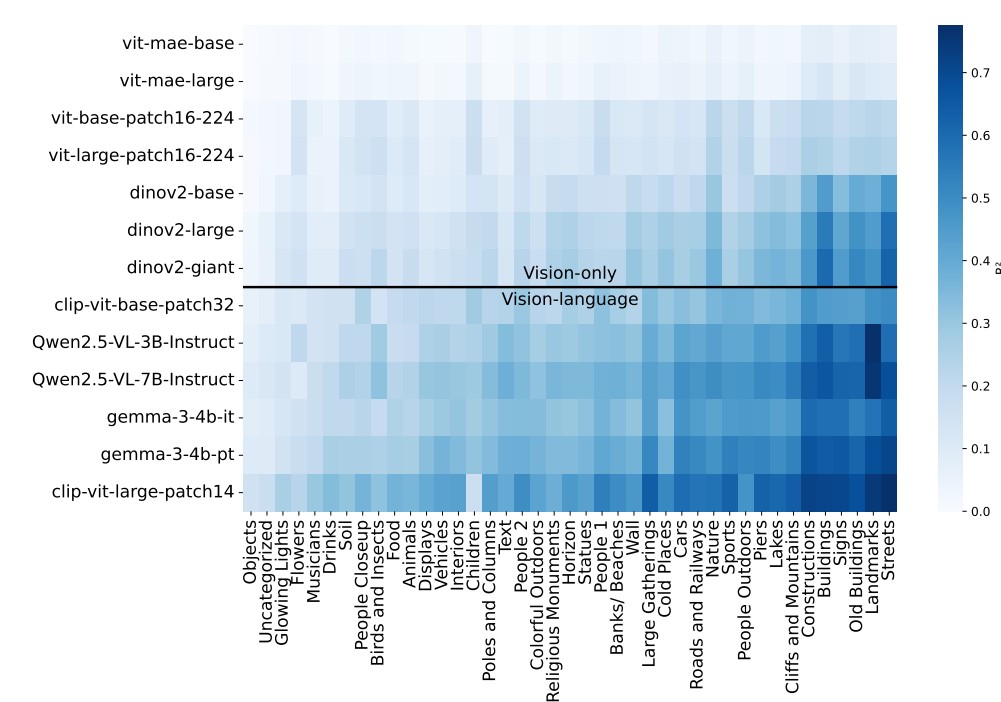

Figure 14: Performance ($R^2$) of each model when using a non-linear probe. The x-axis shows different clusters of the YFCC100M dataset and the Landmarks dataset, while the y-axis shows the models evaluated.

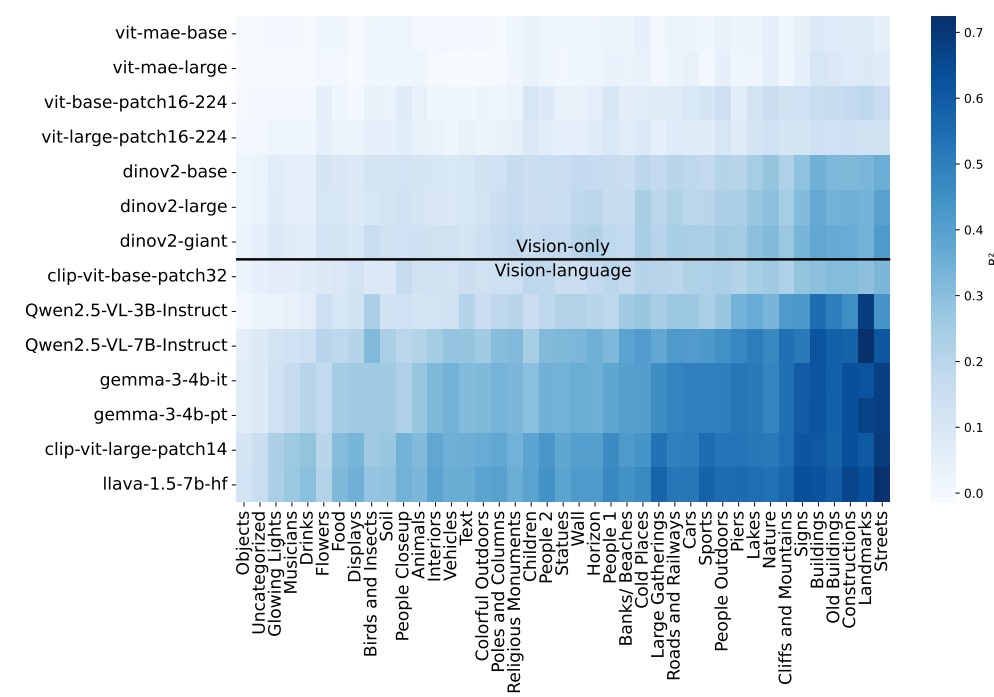

Figure 15: Performance ($R^2$) of each model when using a linear probe and the concatenation of max and min pooling across input tokens as features. The x-axis shows different clusters of the YFCC100M dataset and the Landmarks dataset, while the y-axis shows the models evaluated.

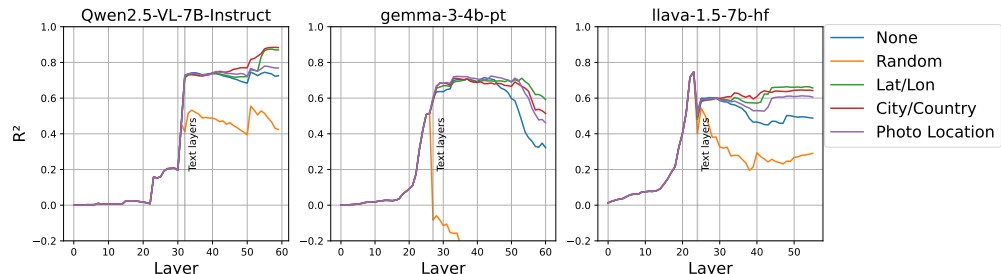

Figure 16: $R^2$ for different prompting strategies on the landmarks dataset. Note how the use of prompts requiring geospatial awareness leads to increased $R^2$ across studied models towards the latter layers.

- **Lat/Lon:** "Guess the latitude and longitude of this image. Answer only with the coordinate tuple (lat, long)"
- **City/Country:** "What country and city was this picture taken in? Answer only with the city and country names."
- **Photo Location:** "Where is this photo?"

It is possible to see that, on average, the more explicit prompts are correlated with higher probing performance, while random and no prompting strategies do not show any significant gain as the information propagates through the network. This further suggests that textual information may help the model retrieve and refine geospatial information from the image representations.

## H  ADDITIONAL REPRESENTATION SWAPPING RESULTS

In this section, we present additional examples of location swap results obtained from the methodology described in Section 4.5. Here, we show that the intervention used the swapping methodology often leads to other changes in the text, e.g., different place names, mixing characteristics from both images, especially if the text generated has very different lengths for the source and target images.

Table 7 shows examples of generated text after swapping fraction $p$ dimensions from the source image with the target image. We observe that some figures have strong features, while others have easily edited geospatial information. For example, with the same fraction of replaced dimensions, we can move the Cologne Cathedral to Paris, but we cannot move the Eiffel Tower to Cologne. Additionally, the St. Peter Cathedral in the Vatican overwrites all other information when used as a target image.

We contrast these results with randomly swapping a fraction $p$ of the embeddings with no regards for geospatial importance (Table 8). It is possible to see that in all cases, by swapping random dimensions, the text generation changes dramatically when compared to our approach, with most samples either keeping their location or having much more added gibberish.

We also evaluate the results quantitatively. For this, we extract a subset of landmarks that are likely known by the model. Namely, we use pictures from the most visited tourist spots from Wikipedia [6], together with the new seven wonders of the world, totaling 51 places. For each of those, we sample five locations from different countries to serve as targets following our approach. We generate the text for each of these samples using the Qwen2.5-VL-3B model and remove the places that the VLM was unable to correctly localize. After these steps, we were left with 208 (source, target) tuples. We apply our steering methodology from eq. 5 and random swapping to this dataset. The results are evaluated regarding the method's efficacy, stability, and success rate.

wXaT.W3
uh9U.W4

We define efficacy as the probability of having the model change its generation for the location from the source to the target location (country, city, or region). Meanwhile, stability refers to the probability of the generated text keeping the original source place name (e.g., Eiffel Tower) after

---

[6]https://en.wikipedia.org/wiki/List_of_most-visited_palaces_and_monuments

Table 7: Results for embedding swapping using varied landmark pictures as source and target images. The changes are very drastic for some image combinations, despite similar methodology when implementing the interventions.

| Source Image | Target Image | $p$ | Generated Text |
|---|---|---|---|
|  |  | 0.40 | The image depicts the Cologne Cathedral, also known as the Cologne Cathedral, located in Cologne, France |
|  |  | 0.40 | The image depicts the Cologne Cathedral, also known as the Cathedral of Notre-Dame de Paris, located in the of Paris, France. |
|  |  | 0.45 | The image depicts the. Peter's Basilica, Vatican City. |
|  |  | 0.50 | The image depicts the Taji Palace, Rome, Italy. |
|  |  | 0.50 | The image depicts the Hagou Basil Mosque, Istanbul, Italy. |

the intervention. Finally, an intervention was successful if both of these conditions applied. These metrics provide a lower bound for the effectiveness of each method as they rely on exact substring matching for identifying whether an intervention was successful or not, and thus, minimal changes such as typos or small amounts of extra gibberish have a big effect on the result.

Table 9 summarizes these metrics for different proportion $p$ values. There is a tradeoff between stability and efficacy, with the former decreasing with higher $p$ while the latter increases. Note how the efficacy of our method is much higher for small $p$ when compared to random, despite showing similar stability. Maximum success rate is achieved at $p = 0.6$, where 11% of the generations successfully included the target location while keeping the source place name.

## I  FINE-TUNING DETAILS

To mitigate the spatial imbalance of our data, which could negatively affect country identification, we constructed a balanced sampling of Google Landmarks across countries. First, we selected a single image for each of the 15,453 labeled landmarks. From these, we retained only those belonging to the 51 countries with over 30 samples, and sampled up to 100 images per country, resulting in a dataset of 3,992 images.

Using this data, we fine-tuned CLIP-large, ViT-large, ViT-MAE-large, and DINOv2 (large and giant) for classification. All models were trained for five epochs with a 70% train, 20% validation, and 10% test splits using an AdamW optimizer, a batch size of 16, and learning rates of $1 \times 10^{-5}$, $2 \times 10^{-5}$,

Table 8: Results for swapping random embeddings using varied landmark pictures as source and target images. Note how, when compared to our approach, the locations are no longer swapped.

| Source Image | Target Image | $p$ | Generated Text |
|---|---|---|---|
|  |  | 0.40 | The image depicts the Cologne Cathedral, also known as the Cologne Cathedral, in Cologne, Germany. |
|  |  | 0.40 | The image depicts the Cologne Cathedral,ptyhek in Cologne, Germany. |
|  |  | 0.45 | The image depicts the Eiffel Tower, a famous landmark in Paris, France. |
|  |  | 0.50 | The image depicts the Taj Mah Mahal, a an iconic monument in India. |
|  |  | 0.50 | The image depicts the Hagia Sophia (Hagia Sophia) in Istanbul, Turkey. |

Table 9: Quantitative results for applying the representation swapping method from Section 4.5 and a random swapping baseline. Qualitatively, higher values of $p$ for both random and our method increase the generation of gibberish and can cause the generated text to mix features from both locations.

| $p$ | Stability | | Efficacy | | Success Rate | |
|---|---|---|---|---|---|---|
| | Our Method | Random | Our Method | Random | Our Method | Random |
| 0.0 | 1.00 | 1.00 | 0.00 | 0.00 | 0.00 | 0.00 |
| 0.1 | 0.97 | 0.98 | 0.00 | 0.00 | 0.00 | 0.00 |
| 0.2 | 0.95 | 0.98 | 0.00 | 0.00 | 0.00 | 0.00 |
| 0.3 | 0.93 | 0.96 | 0.00 | 0.00 | 0.00 | 0.00 |
| 0.4 | 0.73 | 0.76 | 0.01 | 0.00 | 0.01 | 0.00 |
| 0.5 | 0.33 | 0.41 | 0.12 | 0.02 | 0.02 | 0.01 |
| 0.6 | 0.12 | 0.20 | 0.78 | 0.41 | 0.11 | 0.06 |
| 0.7 | 0.05 | 0.10 | 0.93 | 0.83 | 0.04 | 0.07 |
| 0.8 | 0.04 | 0.04 | 0.97 | 0.97 | 0.04 | 0.03 |
| 0.9 | 0.04 | 0.04 | 0.97 | 0.97 | 0.04 | 0.03 |
| 1.0 | 0.03 | 0.03 | 0.97 | 0.97 | 0.03 | 0.03 |

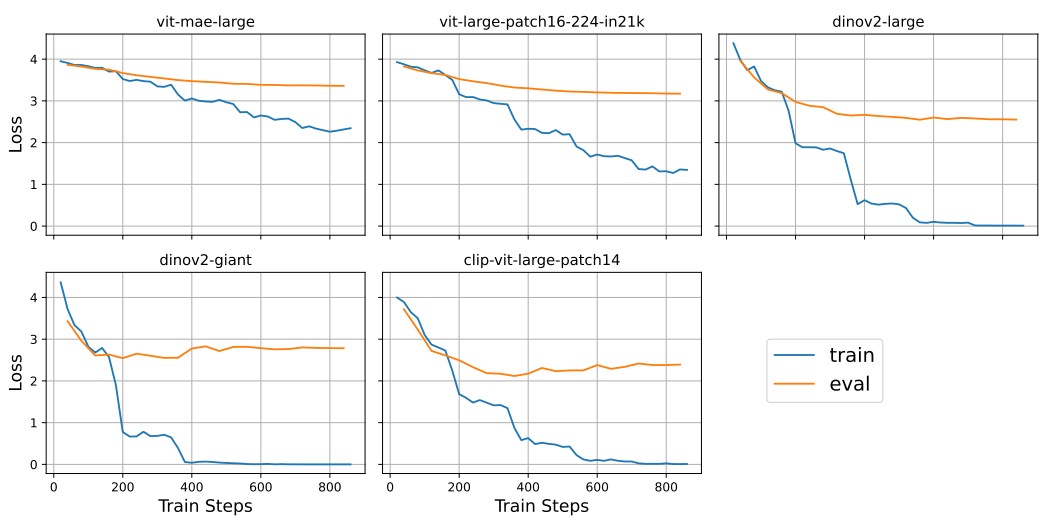

Figure 17: Training and validation loss for model fine-tuning.

$5 \times 10^{-5}$, $1 \times 10^{-4}$, $2 \times 10^{-4}$, and $5 \times 10^{-4}$, with the best chosen based on validation loss. $\beta_1$ and $\beta_2$ were set as 0.9 and 0.999, respectively. Figure 17 shows the training and validation loss for the fine-tuned models. We can see that CLIP and DINOv2 models achieve their best validation loss within 400 training steps, with CLIP's being overall lower, which also translates to better test accuracy.

## J    CONSISTENCY OF INFLUENTIAL WEIGHTS BETWEEN MODELS

Through our probing experiments, we obtain a large set of regression coefficients for latitude and longitude, one pair for each layer and cluster/dataset. Using these coefficients, we investigate to what extent different types of images are represented similarly in the models' latent space. Table 10 shows the average Pearson correlation between each pair of coefficients and the average $R^2$ for a given model on its best performing layer overall.

Table 10: Average correlation ($\rho$) between regression coefficients for different clusters/datasets for each model. The $\pm$ term denotes the 95% confidence interval for the average.

| Model | Average $\rho$ | | Average $R^2$ |
|---|---|---|---|
| | Latitude | Longitude | |
| ViT-MAE-base | $0.142 \pm 0.007$ | $0.115 \pm 0.005$ | $0.035 \pm 0.008$ |
| ViT-MAE-large | $0.143 \pm 0.006$ | $0.111 \pm 0.004$ | $0.051 \pm 0.010$ |
| ViT-base | $0.164 \pm 0.005$ | $0.153 \pm 0.004$ | $0.118 \pm 0.020$ |
| ViT-large | $0.231 \pm 0.006$ | $0.229 \pm 0.005$ | $0.145 \pm 0.020$ |
| DINOv2-base | $0.283 \pm 0.006$ | $0.294 \pm 0.007$ | $0.191 \pm 0.033$ |
| DINOv2-large | $0.312 \pm 0.007$ | $0.291 \pm 0.006$ | $0.236 \pm 0.037$ |
| DINOv2-giant | $0.294 \pm 0.006$ | $0.300 \pm 0.005$ | $0.262 \pm 0.040$ |
| CLIP-ViT-base | $0.410 \pm 0.005$ | $0.500 \pm 0.005$ | $0.278 \pm 0.034$ |
| Qwen2.5-VL-3B | $0.388 \pm 0.005$ | $0.348 \pm 0.004$ | $0.344 \pm 0.047$ |
| Gemma-3-4B-IT | $0.426 \pm 0.005$ | $0.372 \pm 0.005$ | $0.382 \pm 0.046$ |
| Qwen2.5-VL-7B | $0.373 \pm 0.005$ | $0.325 \pm 0.004$ | $0.398 \pm 0.049$ |
| Gemma-3-12B-IT | $0.453 \pm 0.005$ | $0.450 \pm 0.005$ | $0.419 \pm 0.050$ |
| Gemma-3-4B-PT | $0.421 \pm 0.005$ | $0.378 \pm 0.005$ | $0.421 \pm 0.050$ |
| Gemma-3-12B-PT | $0.224 \pm 0.005$ | $0.182 \pm 0.004$ | $0.450 \pm 0.055$ |
| CLIP-ViT-large | $0.548 \pm 0.006$ | $0.587 \pm 0.005$ | $0.482 \pm 0.048$ |
| LLaVA-1.5-7B-HF | $0.573 \pm 0.006$ | $0.612 \pm 0.005$ | $0.510 \pm 0.049$ |

Table 11: Average correlation ($\rho$) between the 40% most important regression coefficients for different clusters/datasets for each model. The $\pm$ term denotes the 95% confidence interval for the average.

| Model | Average $\rho$ | | Average $R^2$ |
|---|---|---|---|
| | Latitude | Longitude | |
| ViT-MAE-base | $0.195 \pm 0.009$ | $0.161 \pm 0.006$ | $0.035 \pm 0.008$ |
| ViT-MAE-large | $0.198 \pm 0.007$ | $0.153 \pm 0.005$ | $0.051 \pm 0.010$ |
| ViT-base | $0.253 \pm 0.007$ | $0.230 \pm 0.005$ | $0.118 \pm 0.020$ |
| ViT-large | $0.349 \pm 0.008$ | $0.351 \pm 0.006$ | $0.145 \pm 0.020$ |
| DINOv2-base | $0.407 \pm 0.008$ | $0.421 \pm 0.008$ | $0.191 \pm 0.033$ |
| DINOv2-large | $0.440 \pm 0.008$ | $0.413 \pm 0.007$ | $0.236 \pm 0.037$ |
| DINOv2-giant | $0.425 \pm 0.007$ | $0.434 \pm 0.007$ | $0.262 \pm 0.040$ |
| CLIP-ViT-base | $0.586 \pm 0.005$ | $0.674 \pm 0.005$ | $0.278 \pm 0.034$ |
| Qwen2.5-VL-3B | $0.548 \pm 0.005$ | $0.498 \pm 0.004$ | $0.344 \pm 0.047$ |
| Gemma-3-4B-IT | $0.455 \pm 0.005$ | $0.399 \pm 0.005$ | $0.382 \pm 0.046$ |
| Qwen2.5-VL-7B | $0.521 \pm 0.005$ | $0.459 \pm 0.004$ | $0.398 \pm 0.049$ |
| Gemma-3-12B-IT | $0.475 \pm 0.005$ | $0.470 \pm 0.004$ | $0.421 \pm 0.049$ |
| Gemma-3-4B-PT | $0.463 \pm 0.005$ | $0.416 \pm 0.005$ | $0.421 \pm 0.050$ |
| Gemma-3-12B-PT | $0.275 \pm 0.005$ | $0.224 \pm 0.005$ | $0.443 \pm 0.055$ |
| CLIP-ViT-large | $0.718 \pm 0.005$ | $0.749 \pm 0.004$ | $0.482 \pm 0.048$ |
| LLaVA-1.5-7B-HF | $0.739 \pm 0.005$ | $0.771 \pm 0.004$ | $0.510 \pm 0.049$ |

It is possible to see that for all models, correlations are positive, suggesting that some spatial information is common for a variety of different image subjects. However, for most of the vision-only models, the correlation is very weak ($\rho < 0.3$). Meanwhile, for the models pretrained with language data, we see that correlations are low ($0.3 \leq \rho < 0.5$) to moderate ($0.5 \leq \rho < 0.7$), indicating some shared neurons for geospatial representation across different image types. These correlations get much higher when we consider only the 40% most important dimensions for predicting coordinates (see Section 4.5), with CLIP and LLaVA achieving values above 0.7, as shown in Table 11.

Table 12: Spearman correlation and Jaccard similarity for the coefficients for the landmarks dataset across five seeds. Jaccard is calculated for the overlap between the top 50% dimensions across runs.

| Model | Spearman $\rho$ | | Jaccard Similarity | |
|---|---|---|---|---|
| | Latitude | Longitude | Latitude | Longitude |
| ViT-MAE-base | 0.78 | 0.72 | 0.52 | 0.47 |
| ViT-MAE-large | 0.76 | 0.71 | 0.49 | 0.46 |
| ViT-base | 0.88 | 0.87 | 0.60 | 0.60 |
| ViT-large | 0.89 | 0.88 | 0.62 | 0.63 |
| DINOv2-base | 0.90 | 0.91 | 0.63 | 0.65 |
| DINOv2-large | 0.93 | 0.93 | 0.71 | 0.71 |
| DINOv2-giant | 0.88 | 0.90 | 0.64 | 0.61 |
| CLIP-ViT-base | 0.89 | 0.91 | 0.64 | 0.66 |
| Qwen2.5-VL-3B | 0.92 | 0.91 | 0.66 | 0.65 |
| Gemma-3-4B-IT | 0.83 | 0.83 | 0.57 | 0.57 |
| Qwen2.5-VL-7B | 0.89 | 0.86 | 0.63 | 0.60 |
| Gemma-3-12B-IT | 0.87 | 0.85 | 0.67 | 0.65 |
| Gemma-3-4B-PT | 0.89 | 0.84 | 0.62 | 0.57 |
| Gemma-3-12B-PT | 0.91 | 0.87 | 0.66 | 0.61 |
| CLIP-ViT-large | 0.94 | 0.92 | 0.72 | 0.69 |
| LLaVA-1.5-7B-HF | 0.93 | 0.94 | 0.68 | 0.70 |

We also check the consistency of the model weights by evaluating how the ranking of the coefficients between embedding dimensions is kept on different seeds. We use Spearman's rank correlation to measure how well the rank of each coefficient is kept across a set of five distinct seeds when compared to our original run. Additionally, we calculate Jaccard similarity for the overlap between the top 50% of coefficients ($p = 0.5$) to measure how consistently relevant the top embedding dimensions are for the ridge regression. Table 12 shows the results for this analysis.

We note that, generally, models have stable top coefficients, regardless of textual supervision. This is illustrated by the high Spearman correlation coefficients and Jaccard similarities, which suggest that the same features are consistently the most useful for extracting geolocation information across runs. Among all models, the ViT-MAE family has the lowest values for both metrics, possibly due to its lower capability of producing meaningful linearly separable geospatial representations from the images.

## K  FULL PROBING RESULTS

Tables 13 and 14 shows the full results for probing experiments with a 95% confidence interval obtained from the mean $R^2$ over five-fold cross validation for each dataset. Confidence intervals are generally low, indicating consistency of our probing approach across different subsets of our data.

Table 13: $R^2$ scores for vision-only models for each cluster and for the landmarks dataset, with columns and rows sorted by average $R^2$ in ascending order. The $\pm$ term indicates the 95% confidence interval.

| | ViT-MAE-base | ViT-MAE-large | ViT-base | ViT-large | DINOv2-base | DINOv2-large | DINOv2-giant |
|---|---|---|---|---|---|---|---|
| **Objects** | $.01_{\pm.01}$ | $.01_{\pm.01}$ | $.01_{\pm.02}$ | $.02_{\pm.01}$ | $.02_{\pm.01}$ | $.05_{\pm.01}$ | $.04_{\pm.01}$ |
| **Uncategorized** | $.01_{\pm.01}$ | $.01_{\pm.01}$ | $.01_{\pm.01}$ | $.03_{\pm.01}$ | $.04_{\pm.01}$ | $.07_{\pm.01}$ | $.08_{\pm.02}$ |
| **Glowing Lights** | $.02_{\pm.01}$ | $.02_{\pm.00}$ | $.03_{\pm.00}$ | $.04_{\pm.01}$ | $.08_{\pm.01}$ | $.12_{\pm.01}$ | $.14_{\pm.02}$ |
| **Musicians** | $.02_{\pm.00}$ | $.02_{\pm.00}$ | $.05_{\pm.01}$ | $.06_{\pm.00}$ | $.06_{\pm.01}$ | $.09_{\pm.01}$ | $.11_{\pm.01}$ |
| **Flowers** | $.02_{\pm.01}$ | $.04_{\pm.01}$ | $.10_{\pm.01}$ | $.15_{\pm.01}$ | $.11_{\pm.01}$ | $.15_{\pm.01}$ | $.17_{\pm.01}$ |
| **Drinks** | $.01_{\pm.01}$ | $.01_{\pm.01}$ | $.05_{\pm.01}$ | $.06_{\pm.01}$ | $.06_{\pm.01}$ | $.09_{\pm.00}$ | $.10_{\pm.00}$ |
| **Soil** | $.02_{\pm.00}$ | $.04_{\pm.01}$ | $.08_{\pm.01}$ | $.12_{\pm.01}$ | $.12_{\pm.01}$ | $.15_{\pm.02}$ | $.18_{\pm.02}$ |
| **Food** | $.03_{\pm.01}$ | $.04_{\pm.00}$ | $.09_{\pm.01}$ | $.11_{\pm.01}$ | $.11_{\pm.01}$ | $.14_{\pm.02}$ | $.15_{\pm.02}$ |
| **Animals** | $.02_{\pm.01}$ | $.04_{\pm.01}$ | $.10_{\pm.02}$ | $.14_{\pm.01}$ | $.13_{\pm.01}$ | $.17_{\pm.02}$ | $.20_{\pm.02}$ |
| **People Closeup** | $.03_{\pm.01}$ | $.06_{\pm.01}$ | $.13_{\pm.01}$ | $.15_{\pm.01}$ | $.13_{\pm.02}$ | $.17_{\pm.01}$ | $.18_{\pm.02}$ |
| **Birds and Insects** | $.02_{\pm.01}$ | $.04_{\pm.01}$ | $.10_{\pm.01}$ | $.17_{\pm.01}$ | $.14_{\pm.01}$ | $.19_{\pm.01}$ | $.23_{\pm.01}$ |
| **Displays** | $.01_{\pm.01}$ | $.02_{\pm.01}$ | $.05_{\pm.01}$ | $.08_{\pm.01}$ | $.09_{\pm.02}$ | $.12_{\pm.01}$ | $.14_{\pm.01}$ |
| **Vehicles** | $.01_{\pm.00}$ | $.03_{\pm.00}$ | $.07_{\pm.00}$ | $.08_{\pm.01}$ | $.10_{\pm.01}$ | $.14_{\pm.00}$ | $.16_{\pm.01}$ |
| **Interiors** | $.01_{\pm.01}$ | $.03_{\pm.01}$ | $.07_{\pm.01}$ | $.09_{\pm.01}$ | $.12_{\pm.01}$ | $.17_{\pm.02}$ | $.18_{\pm.02}$ |
| **Text** | $.01_{\pm.01}$ | $.01_{\pm.01}$ | $.07_{\pm.01}$ | $.08_{\pm.01}$ | $.11_{\pm.01}$ | $.13_{\pm.01}$ | $.15_{\pm.01}$ |
| **Colorful Outdoors** | $.02_{\pm.00}$ | $.03_{\pm.01}$ | $.09_{\pm.01}$ | $.12_{\pm.00}$ | $.14_{\pm.01}$ | $.17_{\pm.01}$ | $.19_{\pm.01}$ |
| **Children** | $.04_{\pm.01}$ | $.08_{\pm.01}$ | $.16_{\pm.01}$ | $.18_{\pm.01}$ | $.15_{\pm.01}$ | $.19_{\pm.01}$ | $.20_{\pm.01}$ |
| **Poles/Columns** | $.01_{\pm.00}$ | $.03_{\pm.01}$ | $.06_{\pm.00}$ | $.10_{\pm.00}$ | $.15_{\pm.01}$ | $.21_{\pm.01}$ | $.23_{\pm.01}$ |
| **Religious Monuments** | $.03_{\pm.01}$ | $.05_{\pm.01}$ | $.08_{\pm.01}$ | $.13_{\pm.01}$ | $.20_{\pm.01}$ | $.24_{\pm.01}$ | $.27_{\pm.02}$ |
| **Statues** | $.03_{\pm.01}$ | $.05_{\pm.01}$ | $.11_{\pm.02}$ | $.13_{\pm.01}$ | $.18_{\pm.01}$ | $.22_{\pm.01}$ | $.25_{\pm.01}$ |
| **Horizon** | $.02_{\pm.01}$ | $.03_{\pm.00}$ | $.10_{\pm.01}$ | $.12_{\pm.00}$ | $.20_{\pm.01}$ | $.25_{\pm.01}$ | $.28_{\pm.01}$ |
| **Wall** | $.03_{\pm.01}$ | $.05_{\pm.00}$ | $.10_{\pm.01}$ | $.13_{\pm.01}$ | $.21_{\pm.01}$ | $.28_{\pm.02}$ | $.31_{\pm.02}$ |
| **People 2** | $.03_{\pm.01}$ | $.05_{\pm.01}$ | $.14_{\pm.01}$ | $.17_{\pm.01}$ | $.17_{\pm.01}$ | $.22_{\pm.01}$ | $.24_{\pm.01}$ |
| **Banks/Beaches** | $.05_{\pm.01}$ | $.06_{\pm.01}$ | $.11_{\pm.01}$ | $.14_{\pm.01}$ | $.16_{\pm.02}$ | $.21_{\pm.01}$ | $.23_{\pm.02}$ |
| **Cold Places** | $.05_{\pm.00}$ | $.07_{\pm.00}$ | $.12_{\pm.01}$ | $.14_{\pm.01}$ | $.23_{\pm.01}$ | $.28_{\pm.01}$ | $.32_{\pm.02}$ |
| **People 1** | $.05_{\pm.01}$ | $.07_{\pm.01}$ | $.18_{\pm.01}$ | $.20_{\pm.01}$ | $.17_{\pm.01}$ | $.22_{\pm.01}$ | $.23_{\pm.02}$ |
| **Large Gatherings** | $.02_{\pm.01}$ | $.05_{\pm.01}$ | $.13_{\pm.01}$ | $.15_{\pm.01}$ | $.19_{\pm.01}$ | $.25_{\pm.01}$ | $.27_{\pm.01}$ |
| **Roads/Railways** | $.04_{\pm.01}$ | $.06_{\pm.01}$ | $.13_{\pm.01}$ | $.15_{\pm.01}$ | $.22_{\pm.01}$ | $.28_{\pm.01}$ | $.31_{\pm.01}$ |
| **Cars** | $.06_{\pm.01}$ | $.07_{\pm.00}$ | $.14_{\pm.01}$ | $.16_{\pm.01}$ | $.19_{\pm.02}$ | $.25_{\pm.01}$ | $.28_{\pm.01}$ |
| **Sports** | $.04_{\pm.01}$ | $.06_{\pm.01}$ | $.17_{\pm.02}$ | $.19_{\pm.01}$ | $.17_{\pm.01}$ | $.25_{\pm.01}$ | $.27_{\pm.01}$ |
| **Lakes** | $.04_{\pm.01}$ | $.05_{\pm.01}$ | $.17_{\pm.02}$ | $.20_{\pm.02}$ | $.28_{\pm.02}$ | $.34_{\pm.02}$ | $.36_{\pm.02}$ |
| **Piers** | $.03_{\pm.01}$ | $.04_{\pm.01}$ | $.12_{\pm.01}$ | $.15_{\pm.01}$ | $.26_{\pm.02}$ | $.32_{\pm.01}$ | $.35_{\pm.01}$ |
| **People Outdoors** | $.06_{\pm.01}$ | $.08_{\pm.01}$ | $.20_{\pm.00}$ | $.22_{\pm.01}$ | $.22_{\pm.02}$ | $.28_{\pm.02}$ | $.32_{\pm.02}$ |
| **Nature** | $.06_{\pm.01}$ | $.08_{\pm.01}$ | $.20_{\pm.01}$ | $.25_{\pm.01}$ | $.30_{\pm.02}$ | $.36_{\pm.02}$ | $.38_{\pm.01}$ |
| **Cliffs/Mountains** | $.04_{\pm.01}$ | $.06_{\pm.01}$ | $.17_{\pm.02}$ | $.20_{\pm.02}$ | $.25_{\pm.02}$ | $.31_{\pm.02}$ | $.35_{\pm.02}$ |
| **Signs** | $.07_{\pm.01}$ | $.09_{\pm.01}$ | $.19_{\pm.01}$ | $.22_{\pm.01}$ | $.33_{\pm.01}$ | $.41_{\pm.01}$ | $.45_{\pm.01}$ |
| **Old Buildings** | $.09_{\pm.00}$ | $.12_{\pm.01}$ | $.21_{\pm.02}$ | $.25_{\pm.01}$ | $.39_{\pm.01}$ | $.47_{\pm.01}$ | $.52_{\pm.01}$ |
| **Constructions** | $.08_{\pm.01}$ | $.11_{\pm.01}$ | $.22_{\pm.02}$ | $.25_{\pm.01}$ | $.36_{\pm.02}$ | $.42_{\pm.01}$ | $.46_{\pm.01}$ |
| **Buildings** | $.09_{\pm.01}$ | $.13_{\pm.01}$ | $.21_{\pm.01}$ | $.25_{\pm.01}$ | $.43_{\pm.03}$ | $.53_{\pm.02}$ | $.59_{\pm.02}$ |
| **Landmarks** | $.09_{\pm.01}$ | $.11_{\pm.02}$ | $.22_{\pm.03}$ | $.26_{\pm.02}$ | $.39_{\pm.03}$ | $.44_{\pm.02}$ | $.48_{\pm.02}$ |
| **Streets** | $.07_{\pm.01}$ | $.10_{\pm.01}$ | $.21_{\pm.01}$ | $.24_{\pm.01}$ | $.45_{\pm.01}$ | $.56_{\pm.01}$ | $.60_{\pm.01}$ |

Table 14: $R^2$ scores for models trained with vision and language data for each cluster and for the landmarks dataset, with columns and rows sorted by average $R^2$ in ascending order. The $\pm$ term indicates the 95% confidence interval.

| | CLIP-base | Qwen2.5-VL-3B | Gemma-3-4B-IT | Qwen2.5-VL-7B | Gemma-3-12B-IT | Gemma-3-4B-PT | Gemma-3-12B-PT | CLIP-large | LLaVA-1.5-7B-HF |
|---|---|---|---|---|---|---|---|---|---|
| **Objects** | $.07_{\pm.01}$ | $.09_{\pm.01}$ | $.10_{\pm.01}$ | $.12_{\pm.01}$ | $.13_{\pm.01}$ | $.12_{\pm.00}$ | $.14_{\pm.01}$ | $.16_{\pm.01}$ | $.17_{\pm.00}$ |
| **Uncategorized** | $.08_{\pm.02}$ | $.11_{\pm.01}$ | $.11_{\pm.01}$ | $.13_{\pm.02}$ | $.13_{\pm.01}$ | $.12_{\pm.01}$ | $.14_{\pm.01}$ | $.18_{\pm.01}$ | $.20_{\pm.02}$ |
| **Glowing Lights** | $.13_{\pm.02}$ | $.12_{\pm.01}$ | $.16_{\pm.01}$ | $.16_{\pm.01}$ | $.18_{\pm.01}$ | $.18_{\pm.01}$ | $.19_{\pm.01}$ | $.27_{\pm.02}$ | $.29_{\pm.01}$ |
| **Musicians** | $.15_{\pm.01}$ | $.15_{\pm.02}$ | $.20_{\pm.02}$ | $.19_{\pm.02}$ | $.21_{\pm.02}$ | $.21_{\pm.02}$ | $.23_{\pm.01}$ | $.31_{\pm.02}$ | $.34_{\pm.02}$ |
| **Flowers** | $.12_{\pm.01}$ | $.22_{\pm.01}$ | $.19_{\pm.01}$ | $.24_{\pm.02}$ | $.20_{\pm.01}$ | $.20_{\pm.01}$ | $.21_{\pm.01}$ | $.24_{\pm.01}$ | $.25_{\pm.01}$ |
| **Drinks** | $.16_{\pm.01}$ | $.16_{\pm.01}$ | $.24_{\pm.02}$ | $.22_{\pm.01}$ | $.27_{\pm.01}$ | $.28_{\pm.01}$ | $.30_{\pm.02}$ | $.34_{\pm.01}$ | $.37_{\pm.01}$ |
| **Soil** | $.16_{\pm.01}$ | $.22_{\pm.02}$ | $.24_{\pm.02}$ | $.28_{\pm.02}$ | $.28_{\pm.02}$ | $.27_{\pm.02}$ | $.30_{\pm.02}$ | $.32_{\pm.02}$ | $.35_{\pm.02}$ |
| **Food** | $.20_{\pm.01}$ | $.19_{\pm.02}$ | $.27_{\pm.02}$ | $.25_{\pm.02}$ | $.29_{\pm.02}$ | $.30_{\pm.03}$ | $.31_{\pm.03}$ | $.37_{\pm.01}$ | $.40_{\pm.01}$ |
| **Animals** | $.22_{\pm.01}$ | $.21_{\pm.01}$ | $.28_{\pm.01}$ | $.26_{\pm.02}$ | $.30_{\pm.01}$ | $.30_{\pm.02}$ | $.32_{\pm.01}$ | $.36_{\pm.02}$ | $.39_{\pm.01}$ |
| **People Closeup** | $.25_{\pm.02}$ | $.22_{\pm.01}$ | $.25_{\pm.01}$ | $.26_{\pm.01}$ | $.28_{\pm.02}$ | $.28_{\pm.01}$ | $.29_{\pm.01}$ | $.38_{\pm.01}$ | $.39_{\pm.01}$ |
| **Birds/Insects** | $.15_{\pm.01}$ | $.30_{\pm.02}$ | $.26_{\pm.01}$ | $.34_{\pm.02}$ | $.27_{\pm.02}$ | $.27_{\pm.02}$ | $.29_{\pm.02}$ | $.33_{\pm.02}$ | $.34_{\pm.02}$ |
| **Displays** | $.23_{\pm.03}$ | $.26_{\pm.02}$ | $.30_{\pm.03}$ | $.32_{\pm.03}$ | $.35_{\pm.03}$ | $.34_{\pm.03}$ | $.37_{\pm.03}$ | $.40_{\pm.03}$ | $.43_{\pm.03}$ |
| **Vehicles** | $.22_{\pm.01}$ | $.27_{\pm.01}$ | $.33_{\pm.02}$ | $.33_{\pm.02}$ | $.39_{\pm.02}$ | $.39_{\pm.02}$ | $.42_{\pm.03}$ | $.43_{\pm.02}$ | $.46_{\pm.02}$ |
| **Interiors** | $.22_{\pm.01}$ | $.24_{\pm.01}$ | $.35_{\pm.02}$ | $.31_{\pm.01}$ | $.38_{\pm.01}$ | $.38_{\pm.02}$ | $.39_{\pm.02}$ | $.44_{\pm.02}$ | $.47_{\pm.02}$ |
| **Text** | $.24_{\pm.02}$ | $.34_{\pm.01}$ | $.35_{\pm.01}$ | $.38_{\pm.02}$ | $.42_{\pm.01}$ | $.40_{\pm.01}$ | $.46_{\pm.02}$ | $.41_{\pm.01}$ | $.44_{\pm.02}$ |
| **Colorful Outdoors** | $.23_{\pm.01}$ | $.28_{\pm.01}$ | $.36_{\pm.00}$ | $.33_{\pm.01}$ | $.39_{\pm.01}$ | $.39_{\pm.00}$ | $.42_{\pm.01}$ | $.43_{\pm.01}$ | $.47_{\pm.01}$ |
| **Children** | $.29_{\pm.01}$ | $.26_{\pm.01}$ | $.31_{\pm.01}$ | $.31_{\pm.01}$ | $.33_{\pm.01}$ | $.34_{\pm.02}$ | $.35_{\pm.01}$ | $.44_{\pm.01}$ | $.46_{\pm.01}$ |
| **Poles/Columns** | $.24_{\pm.02}$ | $.30_{\pm.02}$ | $.34_{\pm.01}$ | $.36_{\pm.02}$ | $.38_{\pm.01}$ | $.37_{\pm.01}$ | $.40_{\pm.02}$ | $.45_{\pm.02}$ | $.48_{\pm.02}$ |
| **Religious Monuments** | $.22_{\pm.02}$ | $.31_{\pm.02}$ | $.34_{\pm.02}$ | $.36_{\pm.02}$ | $.37_{\pm.02}$ | $.37_{\pm.02}$ | $.39_{\pm.01}$ | $.40_{\pm.02}$ | $.43_{\pm.02}$ |
| **Statues** | $.26_{\pm.02}$ | $.33_{\pm.02}$ | $.35_{\pm.03}$ | $.37_{\pm.02}$ | $.39_{\pm.02}$ | $.39_{\pm.02}$ | $.42_{\pm.02}$ | $.44_{\pm.03}$ | $.46_{\pm.02}$ |
| **Horizon** | $.27_{\pm.01}$ | $.31_{\pm.02}$ | $.34_{\pm.02}$ | $.37_{\pm.02}$ | $.37_{\pm.02}$ | $.37_{\pm.02}$ | $.39_{\pm.02}$ | $.46_{\pm.02}$ | $.49_{\pm.02}$ |
| **Wall** | $.24_{\pm.01}$ | $.32_{\pm.02}$ | $.35_{\pm.01}$ | $.38_{\pm.01}$ | $.39_{\pm.01}$ | $.39_{\pm.02}$ | $.42_{\pm.02}$ | $.47_{\pm.02}$ | $.50_{\pm.01}$ |
| **People 2** | $.30_{\pm.01}$ | $.32_{\pm.01}$ | $.36_{\pm.01}$ | $.37_{\pm.01}$ | $.41_{\pm.01}$ | $.42_{\pm.00}$ | $.44_{\pm.01}$ | $.49_{\pm.01}$ | $.53_{\pm.01}$ |
| **Banks/Beaches** | $.28_{\pm.01}$ | $.35_{\pm.02}$ | $.39_{\pm.01}$ | $.41_{\pm.01}$ | $.44_{\pm.01}$ | $.42_{\pm.01}$ | $.46_{\pm.01}$ | $.50_{\pm.01}$ | $.53_{\pm.01}$ |
| **Cold Places** | $.30_{\pm.02}$ | $.35_{\pm.01}$ | $.37_{\pm.02}$ | $.43_{\pm.01}$ | $.42_{\pm.02}$ | $.41_{\pm.02}$ | $.45_{\pm.03}$ | $.51_{\pm.02}$ | $.54_{\pm.02}$ |
| **People 1** | $.32_{\pm.01}$ | $.33_{\pm.02}$ | $.41_{\pm.02}$ | $.40_{\pm.02}$ | $.43_{\pm.01}$ | $.44_{\pm.03}$ | $.47_{\pm.03}$ | $.54_{\pm.01}$ | $.56_{\pm.02}$ |
| **Large Gatherings** | $.34_{\pm.02}$ | $.40_{\pm.03}$ | $.47_{\pm.01}$ | $.45_{\pm.02}$ | $.51_{\pm.01}$ | $.53_{\pm.02}$ | $.56_{\pm.02}$ | $.63_{\pm.02}$ | $.67_{\pm.02}$ |
| **Roads/Railways** | $.31_{\pm.01}$ | $.41_{\pm.02}$ | $.48_{\pm.01}$ | $.48_{\pm.01}$ | $.53_{\pm.01}$ | $.53_{\pm.00}$ | $.56_{\pm.01}$ | $.57_{\pm.01}$ | $.61_{\pm.02}$ |
| **Cars** | $.33_{\pm.01}$ | $.43_{\pm.00}$ | $.50_{\pm.01}$ | $.50_{\pm.01}$ | $.53_{\pm.01}$ | $.56_{\pm.02}$ | $.58_{\pm.01}$ | $.59_{\pm.01}$ | $.62_{\pm.01}$ |
| **Sports** | $.38_{\pm.02}$ | $.42_{\pm.02}$ | $.49_{\pm.02}$ | $.49_{\pm.02}$ | $.54_{\pm.02}$ | $.55_{\pm.03}$ | $.58_{\pm.03}$ | $.62_{\pm.02}$ | $.66_{\pm.02}$ |
| **Lakes** | $.35_{\pm.01}$ | $.44_{\pm.01}$ | $.47_{\pm.01}$ | $.52_{\pm.01}$ | $.53_{\pm.02}$ | $.51_{\pm.02}$ | $.54_{\pm.02}$ | $.61_{\pm.01}$ | $.63_{\pm.01}$ |
| **Piers** | $.34_{\pm.02}$ | $.46_{\pm.01}$ | $.50_{\pm.01}$ | $.52_{\pm.02}$ | $.56_{\pm.01}$ | $.56_{\pm.01}$ | $.59_{\pm.02}$ | $.62_{\pm.02}$ | $.66_{\pm.02}$ |
| **People Outdoors** | $.37_{\pm.02}$ | $.42_{\pm.02}$ | $.50_{\pm.02}$ | $.49_{\pm.03}$ | $.54_{\pm.02}$ | $.56_{\pm.02}$ | $.58_{\pm.02}$ | $.60_{\pm.02}$ | $.63_{\pm.01}$ |
| **Nature** | $.36_{\pm.01}$ | $.44_{\pm.01}$ | $.47_{\pm.01}$ | $.51_{\pm.02}$ | $.50_{\pm.01}$ | $.50_{\pm.02}$ | $.53_{\pm.01}$ | $.57_{\pm.01}$ | $.60_{\pm.01}$ |
| **Cliffs/Mountains** | $.37_{\pm.02}$ | $.48_{\pm.02}$ | $.50_{\pm.02}$ | $.57_{\pm.02}$ | $.56_{\pm.01}$ | $.57_{\pm.01}$ | $.60_{\pm.01}$ | $.63_{\pm.01}$ | $.66_{\pm.01}$ |
| **Signs** | $.43_{\pm.02}$ | $.56_{\pm.00}$ | $.61_{\pm.01}$ | $.61_{\pm.02}$ | $.67_{\pm.00}$ | $.67_{\pm.00}$ | $.72_{\pm.01}$ | $.69_{\pm.01}$ | $.72_{\pm.01}$ |
| **Old Buildings** | $.43_{\pm.01}$ | $.57_{\pm.02}$ | $.58_{\pm.02}$ | $.62_{\pm.02}$ | $.64_{\pm.01}$ | $.64_{\pm.02}$ | $.67_{\pm.01}$ | $.67_{\pm.02}$ | $.70_{\pm.01}$ |
| **Constructions** | $.46_{\pm.01}$ | $.58_{\pm.02}$ | $.62_{\pm.01}$ | $.64_{\pm.01}$ | $.68_{\pm.01}$ | $.68_{\pm.01}$ | $.72_{\pm.01}$ | $.71_{\pm.01}$ | $.74_{\pm.01}$ |
| **Buildings** | $.43_{\pm.02}$ | $.63_{\pm.01}$ | $.62_{\pm.01}$ | $.67_{\pm.02}$ | $.67_{\pm.01}$ | $.67_{\pm.01}$ | $.71_{\pm.01}$ | $.70_{\pm.01}$ | $.73_{\pm.02}$ |
| **Landmarks** | $.46_{\pm.03}$ | $.74_{\pm.02}$ | $.63_{\pm.01}$ | $.75_{\pm.01}$ | $.70_{\pm.01}$ | $.72_{\pm.01}$ | $.75_{\pm.01}$ | $.72_{\pm.01}$ | $.75_{\pm.01}$ |
| **Streets** | $.49_{\pm.01}$ | $.59_{\pm.02}$ | $.67_{\pm.01}$ | $.68_{\pm.01}$ | $.70_{\pm.01}$ | $.72_{\pm.01}$ | $.76_{\pm.01}$ | $.77_{\pm.00}$ | $.80_{\pm.01}$ |

