# OpenReview forum: "Textual Supervision Enhances Geospatial Representations in Vision-Language Models"
_ICLR.cc/2026/Conference — Submitted to ICLR 2026_

### Official Review · Reviewer_uh9U · 2025-10-27

**Soundness:** 2
**Presentation:** 3
**Contribution:** 2
**Rating:** 6
**Confidence:** 4

**Summary:**

The paper explores how geospatial representations emerge in vision-only and vision–language interaction models without explicit geographic supervision. It analyzes how textual supervision influences the implicit spatial understanding of visual models by probing layer-wise embeddings and examining the role of prompts and feature manipulation. The results show that incorporating language signals enhances the encoding and controllability of geospatial semantics, suggesting that multimodal learning provides a stronger foundation for spatial reasoning than purely visual training.

**Strengths:**

The paper shows that CLIP and VLMs outperform larger vision-only models, confirming that language-based supervision induces finer spatial awareness.

The paper carried out several interesting experiments to prove the argument. Such as swapping geospatial features between images predictably alters generated locations while preserving semantics.

The paper is well-structured and easy to follow.

**Weaknesses:**

Without statistical testing, it is unclear whether observed differences are robust across runs.

The mechanism by which textual prompts re-activate spatial signals remains speculative. It is unclear whether this effect is specific to geospatially oriented queries or would also occur with unrelated textual inputs, making the causal relationship between language and spatial activation uncertain.

Section 4.4 defines “top p dimensions ranked by coefficients”, but it remains unclear how the obtained regression coefficients are ensured to be optimal or stable. Without verifying consistency across different initializations or regularization strengths, the ranking of dimensions may reflect run-specific artifacts rather than intrinsic geospatial factors.

Section 4.5 mostly shows qualitative examples, but there’s no quantitative analysis. Personally, I think it should include some measurable results, like how often the location edit actually works or stays consistent, to show the effect isn’t just anecdotal.

**Questions:**

In Section 4.3, does prompt-based improvement depend on the linguistic specificity of the query (e.g., “Where is this photo?” vs. “Guess lat/long”)?

What’s the meaning of “fine-grained”? It is not explicitly defined. Does it refer to coordinate-level precision (e.g., latitude/longitude accuracy) or to semantic granularity (e.g., distinguishing similar landmarks within a region)?

If two images have similar content (e.g., buildings sharing architectural styles) but are located in different countries or even continents, can the probe still recover their coordinates? And in such cases, which would perform better, vision-only or vision-language models? Another interesting observation is that clusters such as Food or People Closeups achieve positive R², even though their visual content seems sort of unrelated to geography from the examples. What cues do models rely on to localize these images?

---

> ### Author Response · Authors · 2025-11-25
> **Response to Reviewer uh9U | Part 1**
>
> We thank the reviewer for the insightful feedback.
>
> ## W1
> > Without statistical testing, it is unclear whether observed differences are robust across runs.
>
> Thank you so much for your comment. ***We now report 95% confidence intervals for the main probing results in Appendix K***, this was calculated from the mean for the $R^2$ obtained through 5-fold cross validation.  The confidence intervals are generally low, suggesting robustness across runs, and, thus, most of the gaps in performance between vision-only and vision-language models are statistically significant.
>
> ## W2 and Q1
> > The mechanism by which textual prompts re-activate spatial signals remains speculative. It is unclear whether this effect is specific to geospatially oriented queries or would also occur with unrelated textual inputs, making the causal relationship between language and spatial activation uncertain.
>
> > In Section 4.3, does prompt-based improvement depend on the linguistic specificity of the query (e.g., “Where is this photo?” vs. “Guess lat/long”)?
>
>
> According to the reviewer's comment, ***we discuss results for five different prompting strategies with varying specificity in Appendix G.*** The prompts and results for two of the models are presented below, with the best $R^2$ in bold:
>
> **Prompts:**
> - **Lat/Lon:** “Guess the latitude and longitude of this image. Answer only with the coordinate tuple (lat, long)”
> - **City/Country:** “What country and city was this picture taken in? Answer only with the city and country names.”
> - **Photo Location:** “Where is this photo?”
> - **Random:** 20 tokens sampled randomly for each picture (consistent across models).
> - **None:** Only the image is given as input.
>
>
> |                          | **Lat/Lon** |               | **City/Country** |               | **Photo Location** |               | **Random** |               | **None**  |               |
> |--------------------------|-----------------------|---------------|----------------------------|---------------|----------------------------|---------------|---------------|---------------|-------------|---------------|
> | **Text Model Depth (%)** | **Qwen2.5-VL-7B**           | **LLaVA-1.5-7B-HF** | **Qwen2.5-VL-7B**                | **LLaVA-1.5-7B-HF** | **Qwen2.5-VL-7B**                | **LLaVA-1.5-7B-HF** | **Qwen2.5-VL-7B**   | **LLaVA-1.5-7B-HF** | **Qwen2.5-VL-7B** | **LLaVA-1.5-7B-HF** |
> | **0 (first text layer)** | 0.71                  | 0.48          | 0.72                       | 0.49          | 0.73                       | 0.51          | 0.41        | 0.40          | 0.73          | 0.52          |
> | **0.25**                 | 0.74                  | 0.59          | 0.73                       | 0.61          | 0.73                       | 0.59          | **0.49**    | **0.32**      | **0.74**      | **0.58**      |
> | **0.5**                  | 0.72                  | 0.57          | 0.76                       | 0.61          | 0.74                       | 0.53          | 0.44        | 0.30          | 0.71          | 0.47          |
> | **0.75**                 | 0.75                  | **0.66**      | 0.84                       | **0.64**      | 0.75                       | **0.61**      | 0.51        | 0.25          | 0.73          | 0.50          |
> | **1 (last text layer)**  | **0.87**              | **0.66**      | **0.88**                   | **0.64**      | **0.77**                   | **0.61**      | 0.42        | 0.29          | 0.73          | 0.49          |
>
> Across all cases, when the prompt relates to image location—even without explicit latitude/longitude—the probes show little to no performance decay across model layers, and often improve. In contrast, with no prompt, performance stabilizes in Qwen but degrades in LLaVA toward the final layers. Using a prompt of 20 random tokens further demonstrates that inadequate prompts substantially reduce representation quality, with both Qwen and LLaVA showing sharp drops in probing performance.

---

> ### Author Response · Authors · 2025-11-25
> **Response to Reviewer uh9U | Part 2**
>
> ## W3
> > Stability of the top-p dimensions ranked by coefficients in Section 4.4.
>
> To address this comment, we ran additional experiments on an additional set of five seeds and evaluated stability by calculating the Spearman correlation between our original coefficients and the coefficients for each of the seeds and the Jaccard similarity between the same set of coefficients for each value of $p$ in $[0.1,0.2,...,1]$. We present the results for $p=0.5$ below.
>
> | **Model**                  | **Latitude Spearman $\rho$** | **Longitude Spearman $\rho$** | **Latitude Jaccard Similarity ($p=0.5$)** | **Longitude Jaccard Similarity ($p=0.5$)** |
> |----------------------------|---------------------------|----------------------------|------------------------------|-------------------------------|
> | ViT-MAE-base           | 0.78                      | 0.72                       | 0.52                         | 0.47                          |
> | ViT-MAE-large          | 0.76                      | 0.71                       | 0.49                         | 0.46                          |
> | ViT-base   | 0.88                      | 0.87                       | 0.60                         | 0.60                          |
> | ViT-large  | 0.89                      | 0.88                       | 0.62                         | 0.63                          |
> | DINOv2-base            | 0.90                      | 0.91                       | 0.63                         | 0.65                          |
> | DINOv2-large           | 0.93                      | 0.93                       | 0.71                         | 0.71                          |
> | DINOv2-giant           | 0.88                      | 0.90                       | 0.64                         | 0.61                          |
> | CLIP-ViT-base  | 0.89                      | 0.91                       | 0.64                         | 0.66                          |
> | Qwen2.5-VL-3B | 0.92                      | 0.91                       | 0.66                         | 0.65                          |
> | Gemma-3-4B-IT          | 0.83                      | 0.83                       | 0.57                         | 0.57                          |
> | Qwen2.5-VL-7B | 0.89                      | 0.86                       | 0.63                         | 0.60                          |
> | Gemma-3-12B-IT         | 0.87                      | 0.85                       | 0.67                         | 0.65                          |
> | Gemma-3-4B-PT          | 0.89                      | 0.84                       | 0.62                         | 0.57                          |
> | Gemma-3-12B-PT         | 0.91                      | 0.87                       | 0.66                         | 0.61                          |
> | CLIP-ViT-large | 0.94                      | 0.92                       | 0.72                         | 0.69                          |
> | LLaVA-1.5-7B-HF        | 0.93                      | 0.94                       | 0.68                         | 0.70                          |
>
> We report this new result in **Appendix J** and **Table 12** in the revised manuscript. We note that, generally, ***all models have stable coefficients***, with high Spearman correlation ($>0.8$) and relatively high Jaccard similarity for $p=0.5$ ($>0.55$), suggesting consistency across runs.

---

> ### Author Response · Authors · 2025-11-25
> **Response to Reviewer uh9U | Part 3**
>
> ## W4
> > Lacking quantitative analysis in Section 4.5.
>
> According to the reviewer's suggestion, in the revised version of the manuscript, ***we add quantitative experiments*** to measure (i) how often the location edit succeeds and (ii) how well non-location content is preserved.
>
> |         | **Stability**  |            | **Efficacy**   |            | **Success Rate** |            |
> |---------|--------------- |------------|----------------|------------|------------------|------------|
> | $p$     | **Our Method** | **Random** | **Our Method** | **Random** | **Our Method**   | **Random** |
> | **0.0** | 1.00           | 1.00       | 0.00           | 0.00       | 0.00             | 0.00       |
> | **0.2** | 0.95           | 0.98       | 0.00           | 0.00       | 0.00             | 0.00       |
> | **0.4** | 0.73           | 0.76       | 0.01           | 0.00       | 0.01             | 0.00       |
> | **0.6** | 0.12           | 0.20       | 0.78           | 0.41       | 0.11             | 0.06       |
> | **0.8** | 0.04           | 0.04       | 0.97           | 0.97       | 0.04             | 0.03       |
> | **1.0** | 0.03           | 0.03       | 0.97           | 0.97       | 0.03             | 0.03       |
>
> **Appendix H** and **Table 9** report edit success rates for Qwen2.5-VL-3B over 208 test examples consisting of tuples (source, target) of famous landmarks. We define *efficacy* as the proportion of examples where the location was changed, *stability* as the proportion where the tourist spot name was mantained, and  *successful* if the location is changed but the tourist spot is mantained. Edit succeeds in up to 11% of cases, which is a lower bound since exact string matching misses minor variations (e.g., Taji Mahal vs Taj Mahal). As the swap proportion $p$ increases, efficacy rises but stability falls, with optimal success at $p=0.6$. Targeted swaps of geospatial embeddings yield higher efficacy and success at lower $p$.
>
> We also report results for Qwen2.5-VL-7B. Larger models are more resistant to single‑token edits, so the intervention only shows impact at higher $p$ values, where random and proposed methods converge at $p=1$. Still, leveraging top dimensions identified by our probes yields a slightly higher overall success rate.
>
> |         | **Stability**  |            | **Efficacy**   |            | **Success Rate** |            |
> |---------|----------------|------------|----------------|------------|------------------|------------|
> | $p$     | **Our Method** | **Random** | **Our Method** | **Random** | **Our Method**   | **Random** |
> | **0.5** | 0.95           | 0.94       | 0.00           | 0.00       | 0.00             | 0.00       |
> | **0.6** | 0.90           | 0.89       | 0.00           | 0.00       | 0.00             | 0.00       |
> | **0.7** | 0.80           | 0.79       | 0.00           | 0.00       | 0.00             | 0.00       |
> | **0.8** | 0.49           | 0.67       | 0.11           | 0.00       | 0.04             | 0.00       |
> | **0.9** | 0.31           | 0.31       | 0.44           | 0.46       | 0.13             | 0.12       |
> | **1.0** | 0.14           | 0.14       | 0.80           | 0.80       | 0.11             | 0.11       |
>
>
> ## Q2
> > What’s the meaning of “fine-grained”? It is not explicitly defined. Does it refer to coordinate-level precision (e.g., latitude/longitude accuracy) or to semantic granularity (e.g., distinguishing similar landmarks within a region)?
>
> Thank you for your comment. To address this comment and avoid such confusion, ***we have removed the term "fine-grained" from the manuscript***. Specifically, (i) we remove the term from the Abstract; and (ii) re-write the sentence explaining the methodology of DINO/DINOv2 in Related Works.

---

> ### Author Response · Authors · 2025-11-25
> **Response to Reviewer uh9U | Part 4**
>
> ## Q3
> ### Q3.1
> > If two images have similar content (e.g., buildings sharing architectural styles) but are located in different countries or even continents, can the probe still recover their coordinates? And in such cases, which would perform better, vision-only or vision-language models?
>
> This is an interesting question. To explore it, we conducted additional ablation studies using original Eiffel Tower photos alongside images of similarly designed towers from different countries. We then tested model predictions with the landmarks dataset probes, and the results are summarized below.
>
> | **Eiffel Tower Location** | **ViT-large**          | **DINOv2-giant** | **CLIP-large**                | **LLaVA**                     | **Gemma-12B**                 |
> |---------------------------|------------------------|------------------|-------------------------------|-------------------------------|-------------------------------|
> | France                    | Atlantic (near Europe) | Portugal         | Atlantic (near Europe)        | Atlantic (near France)        | Algeria                       |
> | Mexico                    | Algeria                | Spain            | Algeria                       | Tunisia                       | Atlantic (near Europe)        |
> | United States             | Atlantic (near Africa) | Morocco          | Atlantic (near North America) | Atlantic (near North America) | Atlantic (near North America) |
> | China                     | Mediterran             | Mediterran       | Turkmenistan                  | Pakistan                      | UAE                           |
> | Iran                      | Italy                  | Algeria          | Libya                         | Libya                         | Atlantic (near Africa)        |
> | Pakistan                  | Greece                 | France           | Spain                         | Italy                         | Libya                         |
>
> The results suggest that ***in such cases it is difficult for models to recover the coordinates for the buildings in other countries.*** Overall, text-based models, like Gemma, LLaVA, and CLIP, seem to have better predictions, even predicting one tower that was located in the USA correctly. Interestingly, for all the photos, DINOv2 predict locations around Europe. We also tested photos for buildings with different architectural styles finding similar patterns in our analysis. In those cases, text-based models also led to increased performance.
>
> ### Q3.2
> > Another interesting observation is that clusters such as Food or People Closeups achieve positive R², even though their visual content seems sort of unrelated to geography from the examples. What cues do models rely on to localize these images?
>
> To investigate what cues models rely on, we collected photos of five food types (American Barbecue, Escargot, Sushi, Lamb Curry, and Prato Feito) and obtained predictions from five models using the food cluster probes (results in the table below). The results are shown in the table below.
>
> | **Food**          | **True Location** | **ViT-large**         | **DINOv2-giant**         | **CLIP-large**                 | **LLaVA**                    | **Gemma-12B**         |
> |-------------------|-------------------|-----------------------|--------------------------|--------------------------------|------------------------------|-----------------------|
> | American Barbecue | United States     | Atlantic (near Europe) | Algeria                  | United States                  | United States                | Northern Mexico       |
> | Escargot      | France            | Portugal              | Morrocco                 | Mediterran (between FR and DZ) | Spain                        | Italy                 |
> | Sushi         | Japan             | Saudi Arabia          | Mediterran (near Turkey) | Turkey                         | Sudan                        | Egypt                 |
> | Lamb Curry    | India             | Iran                  | Iran                     | Pakistan                       | Yemen                        | Iran                  |
> | Prato Feito   | Brazil            | Atlantic (near Africa) | Niger                    | Atlantic (near South America)   | Atlantic (near South America) | Atlantic (near Africa) |
>
> We observe that ***text‑supervised models such as CLIP‑large, LLaVA, and Gemma‑12B produced more geographically consistent predictions***—for instance, American Barbecue was localized near the USA. However, when foods are globally popular and visually similar across regions, such as Sushi, all models mislocalized (e.g., predicting the Middle East). These findings suggest that ***models exploit culturally specific visual cues when available, but struggle when the cues are ambiguous or globally shared.***

---

### Official Review · Reviewer_APe2 · 2025-10-29

**Soundness:** 3
**Presentation:** 3
**Contribution:** 2
**Rating:** 6
**Confidence:** 3

**Summary:**

This paper investigates how different model families — vision-only architectures (e.g., ViT), vision-language models (e.g., CLIP), and large-scale multimodal foundation models (e.g., LLaVA, Qwen, Gemma) — learn and encode geospatial representations.

Through clustering-based analyses of images (people, landmarks, objects) grouped by localizability, the authors demonstrate that textual supervision significantly enhances fine-grained geospatial understanding. The findings suggest that language provides complementary cues for encoding spatial context and highlight multimodal learning as a promising direction for advancing geospatial AI.

**Strengths:**

* Comprehensive evaluation across multiple model families, from vision-only encoders to advanced multimodal foundation models.
* Large-scale dataset construction and systematic analysis covering diverse image categories, enabling meaningful cross-model comparison.
* Empirical evidence supporting the hypothesis that textual supervision improves geospatial representation quality.
* Strong visualization and interpretability results, revealing how current models implicitly capture spatial cues.
* The topic lies at the intersection of interpretability and multimodal representation learning, which is timely and relevant to the ICLR community.

**Weaknesses:**

1. Insufficient discussion on recent visual self-supervised models (e.g., Web-SSL, language-free visual encoders [3]) that may already achieve strong spatial representations without textual supervision.

2. Data imbalance and representational bias — the paper acknowledges that landmark data are unevenly distributed. However, it does not examine whether this imbalance propagates into model performance, particularly for underrepresented regions or low-resource geographies.

3. Lack of temporal analysis — the study focuses on static, long-standing landmarks. It would be valuable to assess model understanding of newer or dynamic landmarks to evaluate the temporal robustness of geospatial representations.

4. Prompt-based enhancement is somewhat obvious — while textual supervision improves performance, the paper does not explore whether finetuning (e.g., SFT) introduces catastrophic forgetting or enhances model specialization.

**Questions:**

1. How does this work conceptually align with or differ from the Platonic Representation Hypothesis [1]? Does geospatial representation follow similar convergence trends across modalities?

2. Could you provide more discussion on how large-scale, language-free visual encoders (e.g., [3]) compare to multimodal ones in terms of spatial representation quality?

3. To what extent does data imbalance affect model geospatial reliability for low-visibility or low-resource regions?

4. Have you analyzed temporal sensitivity — i.e., whether newer landmarks absent from training data are recognized differently by various model families?

5. What would happen if the models were finetuned with explicit geospatial supervision? Would this improve or degrade their general multimodal reasoning ability?

**References**

[1] Huh, M., Cheung, B., Wang, T., & Isola, P. The Platonic Representation Hypothesis. ICML 2025.

[2] He, J., Nie, T., & Ma, W. Geolocation Representation from Large Language Models as Generic Enhancers for Spatio-Temporal Learning. AAAI 2025.

[3] Fan, D., Tong, S., Zhu, J., et al. Scaling Language-Free Visual Representation Learning. ICCV 2025.

[4] Menon, S., & Vondrick, C. Visual Classification via Description from Large Language Models. ICLR 2023.

---

> ### Comment · Reviewer_APe2 · 2025-11-21
>
> I noticed some citations were omitted during sketching my review. Sorry for that. Now I have added [2] and [4] back.

---

> ### Author Response · Authors · 2025-11-25
> **Response to Reviewer APe2 | Part 1**
>
> We thank the reviewer for the insightful feedback and for the thoughtful suggestion of additional related work.
>
> ## W1 and Q2
> > Insufficient discussion on recent visual self-supervised models (e.g., Web-SSL, language-free visual encoders) that may already achieve strong spatial representations without textual supervision.
>
> > Could you provide more discussion on how large-scale, language-free visual encoders compare to multimodal ones in terms of spatial representation quality?
>
> We thank the reviewer for highlighting these works. In response, we ***analyzed recent visual self‑supervised models and found trends consistent with our manuscript***. Specifically, even for these recent models, MAE-based approaches lead to weak representations for our proposed task, followed in performance by DINO and with text-supervised models achieving the best performance. A summary for the landmarks dataset is shown in the table below.
>
> | **Architecture**      | **Objective**                                     | **Parameters** | **Patch Size** | **Train Steps** | **$R^2$** |
> |----------------------------|--------------------------------------------------------|---------------------|---------------------|----------------------|----------------|
> | **MetaCLIP-base**     | Contrastive Image-Text Pretraining | 87M | 16 | 12.8B | 0.53 |
> | **MetaCLIP-large**    |       Contrastive Image-Text Pretraining                                                 | 300M                | 14                  |       12.8B               | 0.70           |
> | **MetaCLIP-huge**     |     Contrastive Image-Text Pretraining                                                   | 600M                | 14                  |    12.8B                  | 0.73           |
> | **Web-SSL DINO-300M** | DINOv2 Loss      | 300M | 14 | 2B | 0.26 |
> | **Web-SSL DINO-1B**   |     DINOv2 Loss                                                   | 1B                  |     14                | 2B                   | 0.39           |
> | **Web-SSL DINO-7B**  |   DINOv2 Loss                                                     | 7B                  |          14           | 8B                   | 0.56           |
> | **Web-SSL MAE-300M**  | Masked Autoencoder      | 300M | 14 | 2B | 0.11 |
> | **Web-SSL MAE-1B**    |      Masked Autoencoder                                                  | 1B                  | 14                  |             2B         | 0.14           |
>
> We add a discussion of these models in **Appendix D** and include full results in **Figure 2**. As noted in our Discussion, scaling vision‑only models improves geospatial representation despite the absence of textual supervision. ***However, even the largest model (Web‑SSL DINO‑7B) performs worse than text‑supervised models such as MetaCLIP***, despite being trained on the same dataset.
>
>
> ## W2 and Q3
> > Data imbalance and representational bias — the paper acknowledges that landmark data are unevenly distributed. However, it does not examine whether this imbalance propagates into model performance, particularly for underrepresented regions or low-resource geographies.
>
> > To what extent does data imbalance affect model geospatial reliability for low-visibility or low-resource regions?
>
> Thank you for motivating the addition of an analysis on data imbalance and model performance. We include this in **Appendix C**, relying primarily on the marked spatial self‑information (SSI) geo‑bias score [1], which captures the tendency to spatially cluster errors. Alongside SSI, we also report two continent‑level performance metrics.
>
> Our evaluation reveals strong spatial bias across all models, reflected in their high SSI scores. Vision‑language models (VLMs) generally outperform vision‑only counterparts (**Figure 12**). A more in depth analysis was also presented for 5 good-performing representative VLM models (**Table 4**). ***Performance is typically higher for images from the Northern Hemisphere, while South America and Oceania show the weakest results***. This discrepancy is illustrated in the table below, which reports median geodesic error for Gemma‑3‑12B‑PT: lowest for Europe and highest for South America and Oceania.
>
> | Model          | Europe    | North America      |   Asia    | South America | Oceania   | Africa    |
> |----------------|-----------|--------------------|-----------|---------------|-----------|-----------|
> | Gemma-3-12B-PT | 752.9 km  | 1768.0 km          | 1546.3 km | 3965.3 km     | 7259.0 km | 1865.9 km |
>
> Our results show that ***current vision and multimodal foundation models display measurable spatial bias, especially against underrepresented regions***. We emphasize this issue in the Discussion to encourage future work on bias‑mitigation strategies that promote fairness in spatial tasks.
>
>
> ### References
> [1] Nemin Wu, Qian Cao, Zhangyu Wang, et al. Torchspatial: A location encoding framework and benchmark for spatial representation learning. *NeurIPS* 2024.

---

> ### Author Response · Authors · 2025-11-25
> **Response to Reviewer APe2 | Part 2**
>
> ## W3 and Q4
> > Lack of temporal analysis — the study focuses on static, long-standing landmarks. It would be valuable to assess model understanding of newer or dynamic landmarks to evaluate the temporal robustness of geospatial representations.
>
> > Have you analyzed temporal sensitivity — i.e., whether newer landmarks absent from training data are recognized differently by various model families?
>
>
> The data used to pretrain these models is usually not disclosed, hence, it is hard to know which information each model had access to before being made available to us. Even new datasets may contain images that were sourced from common crawl or other data sources that may already have been seen by the model during training.
>
> To address the reviewer's comment, we identify a small subset of pictures from the Global Streetscapes dataset [2] which was available only after the release date of most models (we omit Gemma and Qwen as they were released after this dataset). These pictures are from a set of 81 cities from 29 countries. We sample approximately the same number of pictures from each country, obtaining a total of 5k pictures.
>
> Due to the similarity of these pictures to the street cluster from the YFCC dataset, we opt to evaluate their $R^2$ using the probe trained for that cluster, which means that neither the models nor the probe have seen these pictures before. We present the results of probing for this set in the table below:
>
> |           | **ViT-MAE-base** |  **ViT-MAE-large** | **ViT-base** | **ViT-large** | **DINOv2-small** | **CLIP-base** | **DINOv2-base** | **DINOv2-large** | **DINOv2-giant** | **CLIP-large** | **LLaVA-1.5-7B-HF** |
> |-----------|------------------|-------------------|--------------|---------------|------------------|---------------|-----------------|------------------|------------------|----------------|-----------------|
> | **$R^2$** | -0.04             | -0.01              | 0.02         | 0.09          | 0.20             | 0.35          | 0.43            | 0.53             | 0.54             | 0.75           | 0.82            |
>
> Although the models had no prior exposure to Streetscapes images, ***their performance aligns with results on the streets cluster, suggesting they can represent images appearing after their knowledge cut‑off***. While this indicates some generalization, the findings ***may not*** extend to entirely new landmarks with unfamiliar subjects and surroundings. Because it is difficult to determine which data points were included in pretraining, we could not thoroughly assess unseen landmarks.
>
>
> ### References
>
> [2] Yujun Hou, Matias Quintana, Maxim Khomiakov, et al. Global Streetscapes—A comprehensive dataset of 10 million street-level images across 688 cities for urban science and analytics. *ISPRS J. Photogramm. Remote Sens.* 215, 216-238 (2024).
>
> ## W4 and Q5
> >Prompt-based enhancement is somewhat obvious — while textual supervision improves performance, the paper does not explore whether finetuning (e.g., SFT) introduces catastrophic forgetting or enhances model specialization.
>
> >What would happen if the models were finetuned with explicit geospatial supervision? Would this improve or degrade their general multimodal reasoning ability?
>
>
> Thank you for the insightful suggestion. In this work, we focused on spatial representations learned implicitly in the inner layers of models trained with objectives unrelated to geo‑localization, probing directly from latent spaces rather than generated text in VLMs.
>
> To expand on the reviewer’s comment, we discuss supervised approaches such as GeoCLIP [3] and SatCLIP [4], which learn a shared latent space for images and geo‑location data, thereby improving embeddings for geospatial tasks. We also note the recently released GRE Suite [5], which fine‑tunes VLMs with enhanced reasoning chains and reinforcement learning to improve geo‑localization, though its predictions rely on text generation, unlike our latent‑space analysis.
>
> These related works [3, 4, 5] evaluate only geospatial tasks without considering catastrophic forgetting. In contrast, our results (Figure 2) help identify models best suited for integration with methods like GRE Suite in applications focused solely on geo‑localization. ***We agree that studying fine‑tuning effects on both general and geospatial reasoning is an important but underexplored direction, which will be an exciting future direction.***
>
>
>
> ### References
>
> [3] Vicente Vivanco Cepeda, Gaurav Kumar Nayak, and Mubarak Shah. GeoCLIP: Clip-Inspired Alignment between Locations and Images for Effective Worldwide Geo-localization. *NeurIPS* 2023.
>
> [4] Konstantin Klemmer, Esther Rolf, Caleb Robinson, et al. SatCLIP: Global, General-Purpose Location Embeddings with Satellite Imagery. *AAAI* 2025.
>
> [5] Chun Wang, Xiaojun Ye, Xiaoran Pan, et al. GRE Suite: Geo-localization Inference via Fine-Tuned Vision-Language Models and Enhanced Reasoning Chains. *arXiv*, 2505.18700 (2025).

---

> ### Author Response · Authors · 2025-11-25
> **Response to Reviewer APe2 | Part 3**
>
> ## Q1
> > How does this work conceptually align with or differ from the Platonic Representation Hypothesis? Does geospatial representation follow similar convergence trends across modalities?
>
> These are intriguing and fascinating questions. We see many similarities in the convergence trends presented in our results and those results presented by the Platonic Representation Hypothesis (PRH) paper. For the vision-only models, it seems that scaling might be sufficient, but not necessarily efficient, which is a point highlighted by the PRH. In this way, ***we can see textual supervision as enhancing the efficiency for learning geospatial representations.***
>
> Expanding on this discussion, the PRH mention that there might exist a conversion ratio between images and texts in which, for example, a word is worth N pixels for training vision models. Our results seems to show that this conversion ratio proposed in their paper might be dynamic, with different modalities containing different information that are valued differently depending on different downstream tasks.
>
> Finally, in our paper, we do not measure alignment between the top performing methods, but it would be interesting to check if well performing models represent the "world" the same way as analyzed in the PRH. It is a promising future work direction. According to the reviewer's comment, ***we add a discussion between our results and the PRH results in our manuscript.***

---

> > ### Comment · Reviewer_APe2 · 2025-11-25
> >
> > Thanks for your comprehensive response and the PDF update. The results about visual self-supervised models make the claim and conclusion stronger, and the findings about data bias is interesting. Most of my concerns have been addressed.
> >
> > While the discussion about PRH is limited, I believe this paves for future directions about geospatial and other research lines about AI interpretability.
> >
> > I think the paper's presentation and contribution is now significantly better and good enough to present as a poster. Therefore, I'm willing to raise my overall rating to **8**. Good luck!

---

> ### Author Response · Authors · 2025-11-25
> **Thank You!**
>
> Thank you very much for the thoughtful feedback and for raising the rating. We are truly excited!
>
> We acknowledge that our discussion of PRH is limited and agree it points to valuable future directions in geospatial reasoning and AI interpretability. We will consider ways to strengthen this discussion in the revision.
>
> We are grateful once again for the reviewer’s constructive input, which has greatly improved the paper!

---

### Official Review · Reviewer_wXaT · 2025-10-30

**Soundness:** 2
**Presentation:** 2
**Contribution:** 2
**Rating:** 2
**Confidence:** 4

**Summary:**

The paper studies whether vision-only encoders, contrastive VLMs, and instruction-tuned MLLMs implicitly encode geospatial information without explicit geo-supervision. Using layer-wise ridge probes to predict latitude/longitude and reporting R^2, the authors find that models trained with textual supervision show stronger geolocation structure that vision-only models. They also report prompt-conditioning that preserves/boosts geospatial signal in later LLM layers, etc.. Data come from clustered YFCC100M and a sampled Google Landmarks subset with geocell balancing.

**Strengths:**

1. The claim is clear, that is textual supervision systematically strengthens geospatial representations across image types.
2. The authors attempt at geographic balancing via geocells; clusters in YFCC are described and visualized.

**Weaknesses:**

1. The conclusions of this paper are of no value. Its most important takeaway is that multimodal models outperform vision-only models in geospatial representation. This is easy to understand: multimodal models are trained on human knowledge—specifically, text—whereas vision-only models are trained merely to extract visual features without drawing on human knowledge. Under these circumstances, it is only natural that multimodal models would better capture geospatial representation.
2. The authors refer to the ability to identify where a photo was taken as ”geospatial representation,“ which is quite odd. Generally, geospatial representation should refer to the geospatial nature of the image’s own features.
3. Representation-swapping is a single-model, qualitative case study; interesting but not systematically evaluated (stability issues are acknowledged).
4. The prompt that asks for coordinates (“Guess the latitude and longitude…”) may teach the model to surface memorized textual priors rather than reflect purely visual geospatial encoding.

**Questions:**

If you size-match models and hold the image corpus constant while selectively corrupting or removing text supervision during pretraining, do the observed R² and downstream gains persist?

---

> ### Author Response · Authors · 2025-11-25
> **Response to Reviewer wXaT | Part 1**
>
> We thank the reviewer for the insightful feedback.
>
> ## W1
> > The conclusions of this paper are of no value.
>
> While we shared the intuition that multimodal models would outperform vision‑only models in geospatial tasks, no prior work had systematically validated this. ***Our study contributes by empirically confirming this intuition and adding value in three ways***:
> 1. examining the intersection of interpretability and multimodal representation learning to reveal how foundation models structure their latent spaces,
> 2. providing practical guidance for researchers through results such as Figure 2, which highlight model choices for geospatially relevant applications, and
> 3. clarifying where specific information is encoded within model architectures from an interpretability perspective.
>
> Additionally, recent work on large-scale self-supervised vision models -- notably Web-SSL [1] (vision-only model) -- have argued that a systematic vision-only approach can match, or even outperform, vision-language contrastive training. We incorporated Web-SSL into the rebuttal (**Appendix D**) and found that, despite strong performance among vision-only models, it was still overperformed by VLMs of similar size.
>
> Taken together, ***these findings suggest that textual supervision yields substantial gains for geospatial tasks***, while scaling may help narrow the gap between vision‑only and multimodal approaches. We believe our findings offer a clear foundation for future research.
>
>
> ### References
>
> [1] David Fan, Shengbang Tong, Jiachen Zhu, et al. Scaling Language-Free Visual Representation Learning. *ICCV* 2025.
>
> ## W2
> > The authors refer to the ability to identify where a photo was taken as ”geospatial representation,“ which is quite odd. Generally, geospatial representation should refer to the geospatial nature of the image’s own features.
>
> We thank the reviewer for raising this point. To avoid ambiguity, we now ***explicitly define “geospatial” as data linked to specific geographical locations***, and ***“geospatial representation” as latent features learned by neural networks that encode information relevant to downstream geospatial tasks***. This usage is consistent with prior work [2], where street‑view images are treated as sources of geospatial features. We have added this definition to the Introduction and clarified terminology throughout the manuscript.
>
> ### References
>
> [2] Yile Chen, Weiming Huang, Kaiqi Zhao, et al. Self-supervised representation learning for geospatial objects: A survey. *Inf. Fusion*, 103265 (2025).

---

> ### Author Response · Authors · 2025-11-25
> **Response to Reviewer wXaT | Part 2**
>
> ## W3
> > Lacking quantitative analysis in Section 4.5.
>
> Thank you for the suggestion. While we had additional analyses that could not be included in the original submission, we have now added quantitative experiments to evaluate (i) the success rate of location edits and (ii) the preservation of non‑location content.
>
> |         | **Stability**  |            | **Efficacy**   |            | **Success Rate** |            |
> |---------|--------------- |------------|----------------|------------|------------------|------------|
> | $p$     | **Our Method** | **Random** | **Our Method** | **Random** | **Our Method**   | **Random** |
> | **0.0** | 1.00           | 1.00       | 0.00           | 0.00       | 0.00             | 0.00       |
> | **0.2** | 0.95           | 0.98       | 0.00           | 0.00       | 0.00             | 0.00       |
> | **0.4** | 0.73           | 0.76       | 0.01           | 0.00       | 0.01             | 0.00       |
> | **0.6** | 0.12           | 0.20       | 0.78           | 0.41       | 0.11             | 0.06       |
> | **0.8** | 0.04           | 0.04       | 0.97           | 0.97       | 0.04             | 0.03       |
> | **1.0** | 0.03           | 0.03       | 0.97           | 0.97       | 0.03             | 0.03       |
>
> **Appendix H** and **Table 9** report edit success rates for Qwen2.5-VL-3B over 208 test examples consisting of tuples (source, target) of famous landmarks. We define *efficacy* as the proportion of examples where the location was changed, *stability* as the proportion where the tourist spot name was mantained, and  *successful* if the location is changed but the tourist spot is mantained. ***Edit succeeds in up to 11% of cases***, which is a lower bound since exact string matching misses minor variations (e.g., Taji Mahal vs Taj Mahal). As the swap proportion $p$ increases, efficacy rises but stability falls, with optimal success at $p=0.6$. ***Targeted swaps of geospatial embeddings yield higher efficacy and success at lower $p$***.
>
> We also report results for Qwen2.5-VL-7B. Larger models are more resistant to single‑token edits, so the intervention only shows impact at higher $p$ values, where random and proposed methods converge at $p=1$. Still, ***leveraging top dimensions identified by our probes yields a slightly higher overall success rate***.
>
> |         | **Stability**  |            | **Efficacy**   |            | **Success Rate** |            |
> |---------|----------------|------------|----------------|------------|------------------|------------|
> | $p$     | **Our Method** | **Random** | **Our Method** | **Random** | **Our Method**   | **Random** |
> | **0.5** | 0.95           | 0.94       | 0.00           | 0.00       | 0.00             | 0.00       |
> | **0.6** | 0.90           | 0.89       | 0.00           | 0.00       | 0.00             | 0.00       |
> | **0.7** | 0.80           | 0.879      | 0.00           | 0.00       | 0.00             | 0.00       |
> | **0.8** | 0.49           | 0.67       | 0.11           | 0.00       | 0.04             | 0.00       |
> | **0.9** | 0.31           | 0.31       | 0.44           | 0.46       | 0.13             | 0.12       |
> | **1.0** | 0.14           | 0.14       | 0.80           | 0.80       | 0.11             | 0.11       |

---

> ### Author Response · Authors · 2025-11-25
> **Response to Reviewer wXaT | Part 3**
>
> ## W4
> > The prompt that asks for coordinates (“Guess the latitude and longitude…”) may teach the model to surface memorized textual priors rather than reflect purely visual geospatial encoding.
>
> We appreciate the reviewer’s observation. We acknowledge that a prompt explicitly requesting coordinates could encourage the model to rely on memorized textual priors rather than purely visual geospatial encoding. To address this concern, we evaluated five alternative prompting strategies with varying levels of specificity, as detailed in **Appendix G**. By comparing across these prompts, we show that ***our findings are not limited to a single formulation***, and the reported results (with the best $R^2$ highlighted in bold) demonstrate the robustness of our conclusions.
>
> **Prompts:**
> - **Lat/Lon:** “Guess the latitude and longitude of this image. Answer only with the coordinate tuple (lat, long)”
> - **City/Country:** “What country and city was this picture taken in? Answer only with the city and country names.”
> - **Photo Location:** “Where is this photo?”
> - **Random:** 20 tokens sampled randomly for each picture (consistent across models).
> - **None:** Only the image is given as input.
>
>
> |                          | **Lat/Lon** |               | **City/Country** |               | **Photo Location** |               | **Random** |               | **None**  |               |
> |--------------------------|-----------------------|---------------|----------------------------|---------------|----------------------------|---------------|---------------|---------------|-------------|---------------|
> | **Text Model Depth (%)** | **Qwen2.5-VL-7B**           | **LLaVA-1.5-7B-HF** | **Qwen2.5-VL-7B**                | **LLaVA-1.5-7B-HF** | **Qwen2.5-VL-7B**                | **LLaVA-1.5-7B-HF** | **Qwen2.5-VL-7B**   | **LLaVA-1.5-7B-HF** | **Qwen2.5-VL-7B** | **LLaVA-1.5-7B-HF** |
> | **0 (first text layer)** | 0.71                  | 0.48          | 0.72                       | 0.49          | 0.73                       | 0.51          | 0.41        | 0.40          | 0.73          | 0.52          |
> | **0.25**                 | 0.74                  | 0.59          | 0.73                       | 0.61          | 0.73                       | 0.59          | **0.49**    | **0.32**      | **0.74**      | **0.58**      |
> | **0.5**                  | 0.72                  | 0.57          | 0.76                       | 0.61          | 0.74                       | 0.53          | 0.44        | 0.30          | 0.71          | 0.47          |
> | **0.75**                 | 0.75                  | **0.66**      | 0.84                       | **0.64**      | 0.75                       | **0.61**      | 0.51        | 0.25          | 0.73          | 0.50          |
> | **1 (last text layer)**  | **0.87**              | **0.66**      | **0.88**                   | **0.64**      | **0.77**                   | **0.61**      | 0.42        | 0.29          | 0.73          | 0.49          |
>
> When prompts relate to image location, even without explicit lat/lon, the probes show little to no performance decay across layers, and often improve. Without a prompt, performance stabilizes in Qwen but degrades in LLaVA toward the final layers. Using 20 random tokens demonstrates that inadequate prompts sharply reduce representation quality, with both models showing large drops in probing performance.

---

> ### Author Response · Authors · 2025-11-25
> **Response to Reviewer wXaT | Part 4**
>
> ## Q1
> >If you size-match models and hold the image corpus constant while selectively corrupting or removing text supervision during pretraining, do the observed R² and downstream gains persist?
>
> We appreciate the suggestion because differences in data sources and architectural choices can confound results. As fully controlled training from scratch is too costly, we mitigated these factors through alternative measures. To address the reviewer's comment and improve the robustness of our results, ***we compare the MetaCLIP [1] (textual-visual) and Web-SSL [2] (vision-only)*** models, both trained on the same dataset and having comparable compute budget.
>
> | **Architecture**      | **Objective**                                     | **Parameters** | **Patch Size** | **Train Steps** | **$R^2$** |
> |----------------------------|--------------------------------------------------------|---------------------|---------------------|----------------------|----------------|
> | **MetaCLIP-base**     | Contrastive Image-Text Pretraining | 87M | 16 | 12.8B | 0.53 |
> | **MetaCLIP-large**    |       Contrastive Image-Text Pretraining                                                 | 300M                | 14                  |       12.8B               | 0.70           |
> | **MetaCLIP-huge**     |     Contrastive Image-Text Pretraining                                                   | 600M                | 14                  |    12.8B                  | 0.73           |
> | **Web-SSL DINO-300M** | DINOv2 Loss      | 300M | 14 | 2B | 0.26 |
> | **Web-SSL DINO-1B**   |     DINOv2 Loss                                                   | 1B                  |     14                | 2B                   | 0.39           |
> | **Web-SSL DINO-7B**  |   DINOv2 Loss                                                     | 7B                  |          14           | 8B                   | 0.56           |
> | **Web-SSL MAE-300M**  | Masked Autoencoder      | 300M | 14 | 2B | 0.11 |
> | **Web-SSL MAE-1B**    |      Masked Autoencoder                                                  | 1B                  | 14                  |             2B         | 0.14           |
>
> It is seen that the tendencies are similar to those of models trained on different sources, where MAE models consistently fail to represent geospatial information and text supervised models show better performance. Although the MetaCLIP models are trained on more steps, given the difference in the number of parameters, more compute budget is required to train the 7B DINOv2 model for 8B steps comparatively. The fact that MetaCLIP outperforms DINO suggests that, ***even in similar settings, textual supervision enhances the formation of geospatial representations***. For a full discussion, see **Appendix D** of the revised manuscript.
>
> ### References
>
> [3] Hu Xu, Saining Xie, Xiaoqing Ellen Tan, et al. Demystifying CLIP Data. *ICLR* 2024.
>
> [4] David Fan, Shengbang Tong, Jiachen Zhu, et al. Scaling Language-Free Visual Representation Learning. *ICCV* 2025.

---

> ### Author Response · Authors · 2025-11-27
>
> Dear Reviewer,
>
> We hope that many of the concerns have been addressed in our response. If there are any remaining questions, we will be happy to address them!

---

### Official Review · Reviewer_QEj4 · 2025-11-01

**Soundness:** 1
**Presentation:** 1
**Contribution:** 3
**Rating:** 4
**Confidence:** 3

**Summary:**

In this paper, the authors examine the geospatial representation capabilities of pre-trained vision-only foundation models, vision-language models and multimodal LLMs. The authors perform linear probing regression on various layers of the transformers in those pretrained models and analyze their performance in prediction of the longitude and latitude. Using this method, they are able to identify the top features associated with geospatial information and conduct an experiment where they swap that part of the features in text prompts to obtain manipulated model outputs from VLMs. They conclude that textual information is beneficial for geospatial representations in foundation models.

**Strengths:**

1. This paper presents a very interesting and creative research question, i.e. what kind of foundation models are good “geo-guessers”. This is a relatively new and previously unexplored area in understanding the emergent properties of large foundation models.
2. The authors are able to systematically explore and identify layers and dimensions where the models produce latent geospatial information representations. This method makes good contributions to the interpretability of these large models.
3. The experiments shown in Section 4.5 are genuinely surprising and interesting.

**Weaknesses:**

Although I do really like this paper, unfortunately there are a few aspects of this paper that make it not ready for publication right away.
1. The overall story of the paper is very messy and confusing:

    (i) In the introduction section, the authors summarize their research question as: “To what extent do these models internalize global location knowledge as an emergent property of their training and fine-tuning pipelines?” However, the remainder of the paper does not discuss or contain experiments that study this phenomenon as an emergent property. According to [1], “an ability to be emergent if it is not present in smaller models but is present in larger models”. However, this paper does not study model scaling as a factor of this ability.

    (ii) The experiments in Section 4.5, though the most interesting, seem very disconnected to the rest of the paper, since the rest of the paper is more about comparing vision-only encoders to textual-visual models, and Section 4.5 is mainly about steering VLMs. However, Section 4.5 is used as the featured example in Figure 1. This makes the overall flow of the paper extremely confusing.
2. The experimental results are not sufficient to support the main claim: as the title suggests, the authors claim that textual supervision can enhance geospatial representation in those models. However, the comparison conducted in this paper is among various pre-trained models that are trained in completely different settings. The setting differences include model size, training data, and training duration. Given that there are so many confounding factors, I don't think it is reasonable to directly draw the conclusion that textual-visual models are better because they incorporate text data – for example, it is possible that they are better simply because they have a better training data mixture, their model has better architecture, or their model is simply larger. It would be better if the authors compare vision-only models with textual-visual models in a more rigorous setting, i.e. keeping the data source, model size and model architecture the same and only excluding textual information in the vision-only setting.
3. Figure 3 is very confusing – what do these pictures represent in this plot?

Reference:

[1] Wei et al. Emergent Abilities of Large Language Models. 2022.

**Questions:**

It would be great if the authors can answer my questions in the Weakness section.

---

> ### Author Response · Authors · 2025-11-25
> **Response to Reviewer QEj4**
>
> We thank the reviewer for the insightful feedback.
>
> ## W1
>
> ### W1.1
> > The paper does not discuss or contain experiments that study this phenomenon as an emergent property.
>
> Thank you for this insightful comment. We have reframed our research question to focus on the extent to which models internalize global location knowledge during training and fine‑tuning. To avoid overstating our contributions, ***we have removed references to “emergent” and “emergence”*** throughout the manuscript and clarified our analysis accordingly.
>
> ### W1.2
> > Section 4.5 seem very disconnected to the rest of the paper
>
> We agree with the reviewer that the connection was not well explained. To improve coherence, we have ***added a new overview figure*** (**Figure 1**) to illustrate the overall experiment setup. We moved the previous Figure 1 (now **Figure 6**) to its relevant section (**Section 4.5**), ensuring that the discussion is better integrated with the preceding content.
>
> ## W2
> >Compare vision-only models with textual-visual models in a more rigorous setting
>
> We appreciate the suggestion because differences in data sources and architectural choices can confound results. As fully controlled training from scratch is too costly, we mitigated these factors through alternative measures. To address the reviewer's comment and improve the robustness of our results, ***we compare the MetaCLIP [1] (textual-visual) and Web-SSL [2] (vision-only)*** models, both trained on the same dataset and having comparable compute budget.
>
> | **Architecture**      | **Objective**                                     | **Parameters** | **Patch Size** | **Train Steps** | **$R^2$** |
> |----------------------------|--------------------------------------------------------|---------------------|---------------------|----------------------|----------------|
> | **MetaCLIP-base**     | Contrastive Image-Text Pretraining | 87M | 16 | 12.8B | 0.53 |
> | **MetaCLIP-large**    |       Contrastive Image-Text Pretraining                                                 | 300M                | 14                  |       12.8B               | 0.70           |
> | **MetaCLIP-huge**     |     Contrastive Image-Text Pretraining                                                   | 600M                | 14                  |    12.8B                  | 0.73           |
> | **Web-SSL DINO-300M** | DINOv2 Loss      | 300M | 14 | 2B | 0.26 |
> | **Web-SSL DINO-1B**   |     DINOv2 Loss                                                   | 1B                  |     14                | 2B                   | 0.39           |
> | **Web-SSL DINO-7B**  |   DINOv2 Loss                                                     | 7B                  |          14           | 8B                   | 0.56           |
> | **Web-SSL MAE-300M**  | Masked Autoencoder      | 300M | 14 | 2B | 0.11 |
> | **Web-SSL MAE-1B**    |      Masked Autoencoder                                                  | 1B                  | 14                  |             2B         | 0.14           |
>
> The results indicate that the observed tendencies align with those of models trained on different sources: MAE models consistently fail to represent geospatial information and text supervised models achieve stronger performance. Although MetaCLIP models are trained on more steps, given the difference in the number of parameters, more compute budget is required to train the 7B DINOv2 model for 8B steps. The fact that MetaCLIP outperforms DINO suggests that, ***even under comparable settings, textual supervision is important to geospatial representations***.  For a full discussion, see **Appendix D** of the revised manuscript.
>
> ### References
>
> [1] Hu Xu, Saining Xie, Xiaoqing Ellen Tan, et al. Demystifying CLIP Data. *ICLR* 2024.
>
> [2] David Fan, Shengbang Tong, Jiachen Zhu, et al. Scaling Language-Free Visual Representation Learning. *ICCV* 2025.
>
>
> ## W3
> > Figure 3 is confusing
>
> Figure 3 illustrates which clusters are easier or harder for the models to localize. Each picture represents a cluster, and its position corresponds to the model's $R^2$ score for that cluster. The goal is to provide tangible examples of localizability for the studied models. To improve clarity, we have ***revised the figure caption*** and ***updated its reference*** in the main text.

---

### Author Response · Authors · 2025-11-25
**Rebuttal Summary**

We would like to thank all the reviewers for their insightful feedback. In response to the reviews, we conducted new experiments that help us better address the questions and have improved the manuscript's clarity, readability, and flow.

***The strengths of our work lie in the following points, highlighted by the reviewers:***

- The topic is unexplored, timely, and relevant (QEj4,APe2)
- The dataset construction is adequate (wXaT,APe2)
- The experiments and empirical evidence support the claim that textual supervision enhances geospatial representations (QEj4,APe2,uh9U)

***Our responses to the reviews address three main points:***

- Paper re-organization and clarification of scope and terminology
- Strengthening methodological rigor and experimental controls
- Clarification of contributions and positioning within related work

To help track changes more easily, we annotated the margins of the revised manuscript with codes related linked to each review feedback (e.g., ``{reviewer-id}.{itemid}`` format: AAAA.W1 refers to weakness 1 from reviewer AAAA). We also provide a response to every comment.

We are grateful to the reviewers for their feedback, which has improved our work. We have carefully addressed all concerns and are happy to follow up on any remaining points.

Sincerely,

Authors

---

### Meta-Review · Area_Chair_Jndp · 2025-12-28

**Summary:**

This submission received highly diverse scores (6642). The majors concerns raised by reviewers are about technique contributions, clarification of of scope and terminology, and strength of empirical validation. After discussion, major concerns from one of the reviewer were addressed and the reviewer would update the score to 8. Nevertheless the reviewers who assigned negative scores did not respond to the rebuttal. After carefully reading the paper, the review comments and the discussions, I suggest the authors to carefully consider the concerns raised by reviewers and submit their work to the next conference.

**Reviewer Concerns:**

After discussion, major concerns from one of the reviewer were addressed and the reviewer would update the score to 8. Nevertheless the reviewers who assigned negative scores did not respond to the rebuttal.

**Reviewer Scores:**

One reviewer responded to the rebuttal during the discussion phase, but there is no evidence to show all reviewers assigned negative scores would have changed their scores.

---

### Decision · Program_Chairs · 2026-01-26

Reject